# Interplay of RNA m^6^A Modification-Related Geneset in Pan-Cancer

**DOI:** 10.3390/biomedicines12102211

**Published:** 2024-09-27

**Authors:** Boyu Zhang, Yajuan Hao, Haiyan Liu, Jiarun Wu, Lu Lu, Xinfeng Wang, Akhilesh K. Bajpai, Xi Yang

**Affiliations:** 1Department of Hematology, Affiliated Hospital of Nantong University, Nantong 226007, China; 2031110059@stmail.ntu.edu.cn (B.Z.); haiyan1223@126.com (H.L.); 2331110594@stmail.ntu.edu.cn (J.W.); wxf5204079@126.com (X.W.); 2Department of Urology, Shanghai Tenth People’s Hospital, Tongji University, Shanghai 200072, China; haoyajuan1989@126.com; 3Urologic Cancer Institute, School of Medicine, Tongji University, Shanghai 200072, China; 4Department of Genetics, Genomics, and Informatics, University of Tennessee Health Science Center, Memphis, UT 38163, USA; llu@uthsc.edu

**Keywords:** m^6^A, pan-cancer, prognosis, immune cell infiltration, tumor stemness, genomic heterogeneity

## Abstract

**Background**: N^6^-methyladenosine (m^6^A), is the most common modification found in mRNA and lncRNA in higher organisms and plays an important role in physiology and pathology. However, its role in pan-cancer has not been explored. **Results**: A total of 31 m^6^A modification regulators, including 12 writers, 2 erasers, and 17 readers are identified in the current study. The functional analysis of the regulators results in the enrichment of processes, primarily related to RNA modification and metabolism, and the PPI network reveals multiple interactions among the regulators. The mRNA expression analysis reveals a high expression for most of the regulators in pan-cancer. Most of the m^6^A regulators are found to be mutated across the cancers, with *ZC3H13*, *VIRMA,* and *PRRC2A* having a higher frequency rate. Significant correlations of the regulators with clinicopathological parameters, such as age, gender, tumor stage, and grade are identified in pan-cancer. The m^6^A regulators’ expression is found to have significant positive correlations with the miRNAs in pan-cancer. The expression pattern of the m^6^A regulators is able to classify the tumors into different subclusters as well as into high- and low-risk groups. These tumor groups show differential patterns in terms of their immune cell infiltration, tumor stemness score, genomic heterogeneity score, expression of immune regulatory/checkpoint genes, and correlations between the regulators and the drugs. **Conclusions**: Our study provide a comprehensive overview of the functional roles, genetic and epigenetic alterations, and prognostic value of the RNA m^6^A regulators in pan-cancer.

## 1. Introduction

N^6^-methyladenosine (m^6^A) is the most common modification found in mRNA and other kinds of RNA in higher organisms, accounting for 0.1% to 0.4% of all adenine residues [1,2,3]. It typically occurs in the 3′ untranslated region (UTR) and near stop codons in mRNA [4,5]. The m^6^A modification primarily includes m^6^A methylation and m^6^A demethylation.

The m^6^A methylation is defined as the transfer of a methyl group to the N^6^ position of RNA adenosine residues through the action of the methyltransferase complex, using S-adenosylmethionine (SAM) as the methyl donor [6]. The m^6^A demethylation is defined as the removal of m^6^A by the demethylating enzyme transforming it into A, thereby completing the removal of m^6^A [7]. Due to the actions of methyltransferases and demethylases, RNA methylation becomes a dynamic and reversible process. The m^6^A modification could influence the stability and translation efficiency of RNA, which is controlled by three homologous factors, including 12 methyltransferases (defined as “writers”, e.g., *METTL3* [8], *METTL14* [8], *WTAP* [8], *METTL16* [9], *VIRMA* [2], *ZC3H13* [10], *CBLL1* [11], *RBM15* [12], *RBM15B* [12], *METTL5* [13], *TRMT112* [13], and *ZCCHC4* [13]), 2 demethylation enzymes (named “erasers”, e.g., *FTO* [14] and *ALKBH5* [15]), and 17 methylation-binding proteins (also known as “readers”, e.g., *YTHDC1* [16], *YTHDC2* [17], *YTHDF1* [18], *YTHDF2* [19], *YTHDF3* [20], *HNRNPC* [21], *RBMX* [22], *FMR1* [23], *HNRNPA2B1* [24], *IGF2BP1* [25], *IGF2BP2* [25], *IGF2BP3* [25], *PRRC2A* [26], *RBM33* [27], *RBFOX2* [28], *LRPPRC* [29], and *FXR1* [30]). The m^6^A modification reduces mRNA stability [31], increases mRNA translation [18], mediates cytoplasmic liquid–liquid phase separation [32], affectes mRNA splicing [33], and promotes mRNA export from the nucleus [34]. The presence of m^6^A modifications on genes helps to promote/suppress multiple fundamental cellular functions, indicative of its association with many human diseases.

There have been multiple reports about the key role of m^6^A modification in biological processes underlying cancer. The expression of *METTL3* has been shown to be increased in acute myeloid leukemia (AML) patients, suppressing cell differentiation and apoptosis, and promoting cell proliferation through increased translation of *c-MYC*, *BCL2*, and *PTEN*. Further, *METTL3* activates the PI3K/AKT signaling pathway to control cell differentiation and self-renewal in AML [35]. Another study shows that miR-33a inhibits proliferation and promotes differentiation of non-small cell lung cancer cells (NSCLCs) by binding to the 3′UTR of *METTL3* [36], which suggests that *METTL3* may be a novel therapeutic target for NSCLC. Similarly, an increased expression of *METTL3* has been reported in hepatocellular carcinoma (HCC) patients, as well as in in vivo experiments. Furthermore, *METTL3* is found to promote cell growth and migration of HCC, both in vivo and in vitro, and enhances the tumorigenicity, growth, and lung metastasis of liver cancer [37]. The *WTAP* deficiency has been shown to inhibit cell migration, invasion, and tumorigenicity of cholangiocarcinoma (CCA). cDNA microarray and real-time PCR results of CCA have demonstrated that *WTAP* enhances the expression of metastasis-related genes, such as *MMP7*, *MMP28*, and *MUC1* [38].

Other m^6^A regulators also play vital roles in various kinds of cancers, and the same regulator may act contrariwise in different cancers [39]. The eraser *ALKBH5* is downregulated and acts as a tumor suppressor in esophageal squamous cell carcinoma by inhibiting m^6^A/DGCR8-dependent miR-194-2 biogenesis and releasing RAI1 expression, as well as through positive feedback between miR-193a-3p and *ALKBH5*. *ALKBH5* increases in gastric cancer and demethylates lncRNA NEAT1, thus upregulating its expression, and also promotes FAD7 translation in an m^6^A-dependent manner. *ALKBH5* demethylates lncRNA KCNK15-AS1, is decreased in pancreatic cancer, resulting in reduced tumor migration, invasion, and EMT. *ALKBH5* is decreased in non-small cell lung cancer and reduces m^6^A levels on YAP pre-mRNA, leading to suppressed tumor proliferation, migration, invasion, and EMT. However, *ALKBH5* is increased in lung adenocarcinoma and reduces the m^6^A level of FOXM1 mRNA, contributing to an oncogenic role in tumor proliferation and invasion. Another eraser, *FTO,* is elevated, reduces the USP7 mRNA m^6^A level, and promotes the proliferation of non-small cell lung cancer. *FTO* is also elevated, and promotes proliferation and invasion, but inhibits apoptosis by reducing the MZF1 mRNA m^6^A level in lung squamous cell carcinoma. In cervical cancer, *FTO* is upregulated, and promotes proliferation and migration by controlling the m^6^A modification of E2F1 and MYC, also promoting drug resistance by reducing the β-catenin m^6^A level, thereby positively regulating β-catenin expression. *FTO* is increased in acute myeloid leukemia, and regulates the ASB2 and RARA levels by demethylating the m^6^A modification on their RNA, leading to elevated tumor proliferation and reduced apoptosis. *FTO* is also upregulated in breast cancer and mediates the m^6^A demethylation of BNIP3 mRNA, inducing its degradation, leading to an oncogene role in proliferation and metastasis.

The methytransferase *METTL14* is reduced in hepatocellular carcinoma, and promotes pri-miR-126 processing through an m^6^A/DGCR8-dependent manner, thus acting as a tumor suppressor. *METTL14* is also decreased in colorectal cancer and represses tumor progression by promoting SOX4 expression and reducing XIST in an m^6^A-YTHDF2-dependent manner, and by promoting pri-miR-375 processing through an m^6^A/DGCR8-dependent mechanism. *METTL14* also acts as a tumor suppressor in bladder cancer and renal cell carcinoma by inhibiting Notch1 and P2RX6 separately through m^6^A modification. However, *METTL14* is elevated in acute myeloid leukemia by regulating its mRNA targets through m^6^A modification, including MYB, MYC, and SPL1. *KIAA1429* induces GATA3 pre-mRNA methylation and promotes its degradation, leading to the progression of hepatocellular carcinoma.

The m^6^A reader *HNRNPA2B1* upregulates the expression of ACLY and ACC1, leading to the tumor progression of esophageal squamous cell carcinoma. *YTHDF1* promotes FZD7 translation in an m^6^A-dependent manner and is increased to promote the proliferation and metastasis in gastric cancer. *YTHDF1* is also increased and enhanced EIF3C translation using m^6^A modification in ovarian cancer. *YTHDF1* also acts as an oncogene in melanoma by promoting the translation of methylated HINT2 mRNA. *YTHDF2* degrades IL11 and SERPINE2 mRNA, and inhibits tumor growth, vascular density and permeability, and inflammation in hepatocellular carcinoma. *YTHDF2* also increases and promotes 6PGD mRNA translation, leading to lung cancer proliferation. *YTHDF2* is also upregulated in prostate cancer and alters the tumor proliferation and migration using miR-493-3p. *YTHDF3* degrades GAS5 using m^6^A modification and is elevated and enhanced the proliferation and invasion of colorectal cancer. *YTHDC2* increases HIF-1α translation and is increased and promotes the metastasis of colorectal cancer. *IGF2BP2* enhances HMGA2 stability in an m^6^A-independent manner and plays an oncogenic role to promote colorectal cancer migration and invasion. *IGF2BP2* also regulates DANCR stability and enhances pancreatic cancer proliferation and stem cell like properties.

The m^6^A modification has been considered as a key regulator of T cell homeostasis and immune response against bacterial and viral infections. Selectively altering m^6^A levels and other types of immune therapies may be effective strategies for the management of various immune disorders [40,41]. The loss of *YTHDF1* inhibits tumor growth is due to increased infiltration of tumor-specific CD8^+^ T cells in the cancer tissue. The m^6^A modification affectes the turnover and translation of signaling molecules, including that of *MAVS*, *TRAF3*, and *TRAF6*, and thus regulating the production of interferon in antiviral innate immune response [41]. Therefore, m^6^A modification plays important roles in the occurrence and development of cancer, and is thus considered for anti-tumor and anti-viral immune therapies.

In this study, we focuses on the association between m^6^A modification genes and multiple cancers at the genomic, transcriptomic, and proteomic levels, exploring the pan-cancer regulatory mechanism of m^6^A modification.

## 2. Materials and Methods

### 2.1. Geneset Collection, Protein-Protein Interaction (PPI), and Gene Annotation of m^6^A Modifiers

The genes related to the writers, readers, and erasers of m^6^A modification are collected from the literature and are shown in Appendix A. The protein–protein interaction network of the m^6^A modification genes is analyzed and visualized using the STRING database (https://string-db.org/ accessed on 30 May 2024) [42,43]. Furthermore, the Metascape tool (https://metascape.org/ accessed on 30 May 2024) [44] is utilized to explore the Gene Ontology (GO) annotations associated with m^6^A modification genes.

### 2.2. Data Collection and Processing

We download the gene expression data at TPM levels from the GTEx project (https://gtexportal.org/home/ accessed on 30 May 2024) and the Cancer Genome Atlas (TCGA) database (http://cancergenome.nih.gov accessed on 30 May 2024). We collect normal and tumor samples corresponding to 28 tissues, based on the information provided by the GTEx and TCGA databases. To eliminate batch effects, we use the combat algorithm from the sva (v 3.5) package [45]. To analyze the differential expression of genes between normal and tumor samples for each tissue, we employ the Wilcoxon rank sum and signed rank test [46]. In addition, we gather clinical information, mutation data, and miRNA expression data for these 28 types of cancer from the TCGA database [47,48].

### 2.3. Cox Regression Analysis

The univariate Cox regression model in the survival (v 3.2-7) package (https://cran.r-project.org/web/packages/survival/index.html, accessed on 1 March 2024) is used to assess the prognostic significance of each gene by considering their survival time, survival status, and gene expression levels.

### 2.4. Copy Number Variants (CNVs), Single Nucleotide Variations (SNVs), and Methylation Level Analysis

We obtain copy number variants (CNVs), single nucleotide variants (SNVs), and methylation data for multiple cancer types from the GDC (https://portal.gdc.cancer.gov/ accessed on 30 May 2024) database. For each cancer, we calculate the percentages of heterozygous amplifications, homozygous amplifications, heterozygous deletions, and homozygous deletions of m^6^A modification genes. Additionally, we determine the correlation between CNV, methylation status, and RNA expression, as well as the SNV percentage and methylation differences for each cancer. Furthermore, we use the univariate Cox regression model to analyze the survival impact of CNVs and compare the differences between mutant and wild-type cases.

### 2.5. Consensus Cluster Analysis

The R package *ConsensusClusterPlus* (v 1.64.0) is utilized to identify distinct clusters based on the gene expression pattern of each cancer type, using a consensus-clustering approach [49]. The number of clusters is determined based on the area under the curve of the cumulative distribution function and the value of k. To enhance the reliability of the classification outcomes, the classification procedure is repeated 1000 times.

### 2.6. Risk Model Construction

The LASSO regression analysis is used to create a risk model for each cancer, considering factors, such as survival time, survival status, and gene expression levels. This is done using the *glmnet* (v 4.1-7) and *survival* (v 3.5-5) packages [50]. The best model is determined through a 10-fold cross-validation. The *maxstat* (v 0.7-25) package is utilized to calculate the optimal cutoff value for the Risk Score, with the requirement that the minimum group sample size be greater than 25% and the maximum group sample size be less than 75% [51]. Patients are then divided into high- and low-risk groups based on this optimal cutoff value, and further analysis is conducted to examine the prognostic differences between the two groups.

### 2.7. Immune Cell Infiltration, Stemness Features, and Tumor Heterogeneity Analysis

The *CIBERSORT* algorithm from the *IOBR* (v 0.99.9) package is used to elucidate immune cell infiltrations [52]. We collect various scores for tumor dryness, including DNAss (DNA methylation-based), EREG-METHss (epigenetically regulated DNA methylation-based), DMPss (differentially methylated probe-based), ENHss (enhancer element/DNA methylation-based), RNAss (RNA expression-based), and EREG.EXPss (epigenetically regulated RNA expression-based). These scores are calculated using the mRNA expression levels and methylation signatures for each tumor, based on the method outlined in Malta et al.’s research [53]. The tmb function from the maftools package (v 2.18) is used to calculate the tumor mutation burden (TMB) score for each tumor. The inferHeterogeneity function, also from the maftools package, is used to calculate the MATH score (mutant-allele tumor heterogeneity) [54]. The microsatellite instability (MSI) score is calculated based on the method described in Bonneville et al.’s study [55]. Additionally, the neoantigen (NEO) score, purity, ploidy, homologous recombination deficiency (HRD), and loss of heterozygosity (LOH) scores are calculated based on the method outlined in Thorsson et al.’s study [56]. We also calculate the correlation between immune cell infiltrations (or stemness features and tumor heterogeneity) and m^6^A modification genes. Furthermore, we identify the different immune cells (or stemness features and tumor heterogeneity) among clusters, as well as between the high- and low-risk groups.

### 2.8. Immune Regulator and Immune Checkpoint Gene Analysis

The immune regulatory and checkpoint genes are gathered from the study conducted by Hu et al. in 2021 [57]. These genes encompass chemokine receptors, MHC genes, immunoinhibitors, immunestimulators, inhibitory, and stimulatory genes. We perform a calculation to ascertain the correlations between the immune regulator and immune checkpoint genes with m^6^A modification genes. Additionally, we identify distinct immune regulator and immune checkpoint genes among clusters, as well as between the high- and low-risk groups.

### 2.9. Immune Score and miRNA Analysis

The ESTIMATE package (v 1.0.13) is utilized to compute the immune scores for each patient according to the gene expression profiles specific to their respective cancers [58]. Subsequently, we determine the correlations between immune scores and m^6^A modification genes. The miRNA expression data of each tumor patient are acquired from the TCGA database. Then, we compute the correlation coefficient between the miRNAs and m^6^A modification genes.

### 2.10. Drug Prediction

The oncoPredict (v 0.2) package is utilized to analyze the IC50 value of each drug from the Genomics of Drug Sensitivity in Cancer (GDSC) database (https://www.cancerrxgene.org/ accessed on 30 May 2024) [59,60] for each patient, based on their gene expression levels. We select the top 11 drugs with the lowest IC50 values for each cancer type. Subsequently, we compute the differences in the selected drugs between the high- and low-risk groups.

### 2.11. Statistical Analyses

The R (v 4.2.3) software packages are used for data processing, statistical analysis, and plotting. The Pearson’s correlation coefficient is used to calculate the correlation between two continuous variables. The *chi-squared* test is conducted to compare the categorical variables, and the Wilcoxon rank-sum test or *t*-test is used to compare continuous variables.

### 2.12. Identifying m^6^A-Related Biomarkers by Three Machine Learning Algorithms

Three machine-learning algorithms, including LASSO, support vector machine-recursive feature elimination (SVM–RFE) [61], and random forest (RF) [62] algorithms are used to identify the m^6^A-related biomarkers in each cancer type based on the RNA expression and CNV. We introduce the LASSO algorithm above. We use the SVM algorithm to train a model based on the RNA or CNV data [63], and then the SVM–RFE algorithm is employed to iteratively refine the feature set by eliminating the least significant features and enhancing the model’s predictive accuracy [61]. The RF algorithm is used to rank the importance of genes [62]. The genes with a relative importance score above 0.25 are considered significant. The biomarkers are identified by the intersection results of LASSO, SVM-RFE, and RF methods.

## 3. Results

### 3.1. Expression Pattern and Functional Analysis of RNA m^6^A Modification Regulators in Pan-Cancer and Normal Tissues

In the current study, we screen a total of 31 RNA m^6^A modification regulators including 12 writers, 2 erasers, and 17 readers from the published literatures (Appendix A). The m^6^A modification regulators are studied across 28 cancer types from the TCGA database (Appendix A) and 45 normal tissue types from both the TCGA and GTEx databases (Appendix A). The PPI network analysis suggests multiple interactions among the regulators, except for PRRC2A, RBM33, LRPPRC, and RBFOX2 (Figure 1A). The GO analysis of the regulators reveals the enrichment of processes mainly associated with mRNA metabolism, RNA modification, and regulation of mRNA stability, and translational and stem cell population maintenance (Figure 1B). To investigate the expression level of m^6^A modification regulators in normal tissues, we analyze the protein expression of the regulators in 45 types of normal tissues from TCGA and GTEx. Our results find that most of the regulators are highly expressed across all normal tissues except IGF2BP1, IGF2BP3, and RBM15B (Figure 1C). Furthermore, we analyze the mRNA expression of the 31 regulators across 28 cancer types in the TCGA database, as well as compare them with TCGA and GTEx normal cohorts. Our results reveal that the m^6^A regulators are highly expressed in most of the cancer types, such as CHOL, ESCA, GBM, HNSC, LGG, LAML, PAAD, and STAD, while they are decreased in a few tumor types, including ACC, KICH, OV, THCA, and UCS; *METTL3*, *YTHDC1*, *YTHDC2*, *RBM33,* and *RBFOX2* are preferably lower expressed, while *CBLL1*, *RBM15*, *RBM15B*, *METTL5*, *TRMT112*, *YTHDF1*, *YTHDF2*, *YTHDF3*, *HNRNPC*, *HNRNPA2B1*, *IGF2BP1*, *IGF2BP2*, *IGF2BP3*, *PRRC2A*, *LRPPRC* and *FXR1* are highly expressed in most tumor types (Figure 1D). Appendix A represents the expression differences in *IGF2BP2* and *IGF2BP3* between their normal and tumor tissues, respectively.

### 3.2. Correlations among RNA m^6^A Modification Regulators in Pan-Cancer

To explore the relationship among the RNA m^6^A regulators in pan-cancer, we calculate the correlation coefficients among the 31 regulators across the 28 cancer types at the mRNA level (Appendix A). Interestingly, our results revealed that overall-positive correlations are more common than negative correlations, particularly in ACC, KIRP, KIRC, KICH, PCPG, THCA, and UCEC. *METTL5* and *TRMT112* are negatively related to other genes in some cancer types, especially in OV, PRAD, BRCA, and GBM.

### 3.3. Overall Survival (OS) Status Based on the Expression of RNA m^6^A Modification Regulators in Pan-Cancer

We analyze the overall survival (OS) status of 28 TCGA cancer types based on the expression of 31 RNA m^6^A regulators and find the genes to have prognostic significance in different cancer types. Some are found to have a good OS advantage, such as *METTL3* in LIHC, LAML, ACC, and KICH; *METTL14* in LGG and LAML; *WTAP* in LGG, CESC, and LIHC; *VIRMA* in LGG, CESC, and LIHC; *ZC3H13* in LAML; *CBLL1* in LGG and KICH; *RBM15* in LGG, LUAD, LIHC, THCA, LAML, ACC, and KICH; *RBM15B* in LIHC and ACC; *METTL5* in LUAD, KIRP, HNSC, LIHC, PAAD, LAML, ACC, and KICH; *TRMT112* in LGG, KIRP, HNSC, LIHC, LAML, ACC, and KICH; *ZCCHC4* in LGG, LIHC, LAML, and KICH; *FTO* in STAD, BLCA, and LAML; *ALKBH5* in LGG, GBM, BLCA, LAML, and KICH; *YTHDC2* in LGG; and *YTHDF1* in LGG, LIHC, THCA, and LAML (Appendix A). A few genes have a bad OS disadvantage, such as *METTL3* in PAAD; *METTL14* in KIRC, SKCM, and READ; *METTL16* in CESC and PAAD; *WTAP* in SKCM; *VIRMA* in SKCM; *ZC3H13* in KIRC; *CBLL1* in KIRC; *RBM15* in KIRC and READ; *RBM15B* in KIRC; *METTL5* in OV; *TRMT112* in OV; *ZCCHC4* in KIRC and SKCM; *FTO* in KIRC; *ALKBH5* in ESCA, OV, and PAAD; *YTHDC1* in KIRC; *YTHDC2* in COAD, KIRC, and READ; and *YTHDF1* in READ (Appendix A).

### 3.4. Clinical Significance of RNA m^6^A Modification Regulators in Pan-Cancer

To investigate the relationship between the expression levels of m^6^A modification regulators and the clinicopathological features, such as age, gender, grade, stage, tumor T stage, lymph node metastasis, and distant metastasis in pan-cancer, we compare the mRNA expression of the 31 m^6^A regulators in the TCGA pan-cancer cohort across different clinical features. We list the patients’ number distribution of clinicopathological characteristics in each cancer cohort in Appendix A.

Most of the genes are found to be affected by age in KIRP, BRCA, ESCA, LUSC, PAAD, and THCA (Figure 2A). Furthermore, the expression of some genes varies significantly by the patients’ gender in a few cancer types, such as HNSC, KIRP, KIRC, and LIHC (Figure 2B). Similarly, the expression of a few regulators is affected by the patients’ tumor grade, in cancers such as HNSC, KIRC, LGG, LIHC, PAAD, and UCEC; *LRPPRC* and *TRMT112* are affected by tumor grade in most of the cancers (Figure 2C). As shown in Figure 2D, the expression of the regulators significantly varies based on tumor stage in a few cancers, including KIRP, KIRC, KICH, LIHC, OV, THCA, and TGCT; *IGF2BP2* in KIRC and THCA; *IGF2BP3* in KIRP, KIRC, and UCEC; and *LRPPRC* and *METTL14* in KIRC are the ones that are the most varied. Additionally, the expression of most of the regulators influences the tumor T feature of KIRC, LIHC, PRAD, STAD and THCA, especially *IGF2BP1* in BRCA; *IGF2BP2*, *IGF2BP3*, *LRPPRC*, and *METTL14* in KIRC; and *IGF2BP3* in KIRP (Figure 2E). The expression of some of the regulators is found to be affected by the tumor N feature in COAD, HNSC, KIRP, KICH, LUSC, PRAD and THCA, especially *ALKBH5* and *IGF2BP2* in THCA, and *FXR1* and *IGF2BP3* in KIRP (Figure 2F); and the tumor M feature in ACC, KIRC, and LUAD, especially *IGF2BP2* and *IGF2BP3* in KIRC, and *IGF2BP3* in KIRP (Figure 2G).

As shown in Appendix A, as a case study, the expression of *IGF2BP2* in some tumor types is positively correlated with the patients’ age, i.e., the older the patient, the higher the expression, while its expression in a few tumor types is negatively correlated with age, i.e., the younger the patient, the higher the gene expression. Another gene, *YTHDF1*, has higher expression in males than in females in some cancers, while this trend is the opposite for a few other cancers (Appendix A). When we look at *IGF2BP2’s* expression pattern, it is found to vary significantly among G1, G2, G3, and G4 grades in a few cancer types (Appendix A). The expression of *IGF2BP3* is found to be significantly different across stages I–IV in a few cancer types (Appendix A), and across T1–T4 groups (Appendix A). As shown in Appendix A, the expression pattern of a representative gene, *IGF2BP3*, varies significantly across N0–N3 groups and between M0 and M1 groups in some cancer types, respectively.

### 3.5. Correlation between Genetic or Epigenetic Alterations and Expression Levels of m^6^A Regulators in Pan-Cancer

We analyze the correlations of copy number variation (CNV), single nucleotide variation (SNV), methylation levels or genomic variations, including missense mutation, nonsense mutation, frame-shift deletion, frame-shift insertion, in-frame deletion, and in-frame insertion, with the expression levels of the 31 RNA m^6^A modification regulators and overall survival of the patients in pan-cancer.

The CNVs are mainly categorized into heterozygous and homologous amplifications and deletions. The deletions are found to be more common in *METTL16*, *ALKBH5*, *RBM15B*, *METTL14*, *ZC3H13*, *YTHDF2*, *WTAP*, *ZCCHC4*, *RBFOX2*, *YTHDC1*, *YTHDC2*, *RBM15*, *METTL3*, *HNRNPC*, *FTO*, *TRMT112*, and *RBMX*, whereas amplifications are more common in *FMR1*, *METTL5*, *PRRC2A*, *IGF2BP1*, *LRPPRC*, *IGF2BP2*, *FXR1*, *RBM33*, *CBLL1*, *YTHDF3*, *HNRNPA2B1*, *IGF2BP3*, *VIRMA*, and *YTHDF1* in most tumor types. However, a few cancer types, such as THCA, PRAD, LAML, and LGG have a lower percentage of CNVs than the others (Appendix A). Furthermore, we find a positive correlation between CNVs and mRNA expression levels for many of the regulators in most tumor types, especially in BRCA, LUSC, OV, LUAD, HNSC, BLCA, COAD, LIHC, CESC, STAD, SKCM, ESCA, and LGG, and *YTHDF1* in BRCA is the strongest among them (Figure 3A).

Many of the m^6^A regulators have a high SNV mutation frequency in most tumor types, especially in UCEC, SKCM, COAD, and STAD, furthermore, *ZC3H3* in UCEC and *PRRC2A* in SKCM are the top two whose mutation frequency is above 50% (Figure 3B).

Next, we analyze the correlation of patients’ RNA m^6^A regulators’ geneset CNVs with four survival types, i.e., overall survival (OS), progression free survival (PFS), disease specific survival (DSS), and disease-free interval (DFI). Only a small proportion of tumors are found to be affected. For example, the CNVs of m^6^A regulators are correlated with the OS, PFS, and DSS of LGG and GBM patients, while they are correlated with only the DSS of THCA patients. The CNVs of RNA m^6^A regulators correlated with the DFI of PCPG patients (Figure 3C). Furthermore, we analyze the association between SNV mutations in m^6^A regulators and patient survival. The SNV mutations in these regulators are significantly associated with the OS and DSS of UCEC and BLCA patients, the PFS of UCEC, THCA, LIHC, and BLCA patients, and the DFI of LIHC patients (Figure 3D).

We also analyze the genomic variations, including the missense mutation, nonsense mutation, frame-shift deletion, frame-shift insertion, in-frame deletion, and in-frame insertion of 31 RNA m^6^A regulators across 28 types of TCGA pan-cancer cohorts. Most of them are found to be mutated in different cancers, especially in UCEC, COAD, STAD, BLCA, CESC, LUAD, LUSC, READ, and UCS (Figure 3E). *ZC3H13*, *VIRMA*, and *PRRC2A* almost have a higher mutation frequency rate in all cancer types (Figure 3E). Furthermore, we assess different genetic alteration types for *ZC3H13*. Our results reveal that *ZC3H13* has all the six types of alterations, while missense mutation is found to be the most frequent one (Appendix A).

The methylation levels of RNA m^6^A regulators, such as *METTL5*, *YTHDC1*, *TRMT112*, *RBM15B*, *ZCCHC4*, *FMR1*, and *HNRNPA2B1* are found to be decreased in LUSC, KIRC, LIHC, COAD, BRCA, PRAD, LUAD, KIRP, and BLCA; however, the methylation levels of *FTO*, *IGF2BP3*, *IGF2BP2*, *IGF2BP1*, *WTAP*, and *RBM33* are increased in BRCA and LUAD, whereas the levels of *HNRNPC*, *VIRMA*, *LRPPRC*, *ZC3H13,* and *YTHDF2* increased in KIRP (Figure 3F). Interestingly, the methylation levels of most of the RNA m^6^A regulators are negatively correlated with their mRNA expression in different cancer types, especially *METTL5*, *YTHDC1*, *VIRMA*, *FXR1*, *YTHDF2*, and *CBLL1* in TGCT; *IGF2BP3* and *IGF2BP2* in LUSC, HNSC, and PAAD; and *IGF2BP2* in THCA and LAML; and *YTHDF3* in UCS (Figure 3G).

### 3.6. Distinguishing Different Clusters of RNA m^6^A Regulators in TCGA Pan-Cancer Cohort

The R package *ConsensusClusterPlus* is applied to classify patients by the similarity of the m^6^A regulators’ level into two to four different categories, that is, C1, C2, C3 and C4 in the TCGA pan-cancer cohort (Figure 4A). The discrepancy of the RNA m^6^A regulators’ level is found to be significant among C1–C4 groups in the TCGA pan-cancer cohort (Figure 4B). Figure 4C shows the comparison of the differential expression of a representative gene, *TRMT112* across the clusters in each tumor.

### 3.7. Evaluating the Prognostic Value of RNA m^6^A Regulators in TCGA Pan-Cancer Cohort

The LASSO cox regression algorithm is applied to these RNA m^6^A regulators in the TCGA pan-cancer cohort. A few candidate genes are screened out as they are considered to be prognosis-related in different cancers (Appendix A and Figure 5A). We divide the patients into high- and low-risk groups according to the LASSO results. Patients in the high-risk group have a worse prognosis than the patients in the low-risk group in the TCGA pan-cancer cohort (Figure 5B). For example, *IGF2BP3* is found to be greater in high-risk than in low-risk group of ACC, BLCA, CESC, KIRP, KICH, LGG, LUAD, LIHC, LAML, and PAAD; however, the trend is the opposite for READ (Figure 5C). Figure 5D shows the differential expression of RNA m^6^A regulators between the high- and low-risk groups in each tumor. Overall, the genes are found to be increased in the high-risk than in the low-risk group of ACC, BLCA, CESC, KIRP, KICH, LGG, LUAD, LIHC, and PAAD, while decreased in the high-risk compared to the low-risk group of KIRC and READ.

### 3.8. Correlation between Immune Infiltrating Score and RNA m^6^A Regulators in TCGA Pan-Cancer Cohort

Next, we analyze the correlation between the immune infiltrating score with the expression of RNA m^6^A regulators in the TCGA pan-cancer cohort. The genes are negatively related to the immune score in most cancer types except in COAD, KICH, LGG, PAAD, and READ (Appendix A). *IGF2BP2*, *IGF2BP3*, *RBM15*, *WTAP*, and *YTHDC2* are positively correlated to the immune score in a few cancer types (Appendix A). Appendix A shows the most significant correlation of m^6^A regulators with immune score in cancers. For instance, the genes negatively related to the immune score are *YTHDC2* in ACC; *METTL3* in BLCA and KIRP; *RBMX* in BRCA, GBM, LAML, OV, and TGCT; *LRPPRC* in CESC, ESCA, KIRC, LUAD, and SKCM; *METTL5* in HNSC and LUSC; *RBFOX2* in LGG; *ZCCHC4* in PCPG; *YTHDF1* in STAD; *ALKBH5* in THCA; *FXR1* in UCEC; and *PRRC2A* in UCS. The positively correlated genes with immune score include *ALKBH5* in COAD; *WTAP* in CHOL, LIHC, and PAAD; *IGF2BP1* in KICH; *IGF2BP2* in PRAD; and *RBFOX2* in READ (Appendix A).

### 3.9. Association of Tumor Microenvironment (TME) Infiltrating Cells with RNA m^6^A Regulators in Pan-Cancer by CIBERSORT

To study the TME-infiltrating immune cells in TCGA pan-cancer, we use the CIBERSORT method to determine the cell types. Our results show that there are 22 types of tumor-infiltrated immune cells, including subtypes of B cells, T cells, NK cells, macrophages, dendritic cells, mast cells, monocytes, eosinophils, neutrophils, and plasma cells.

As shown in Appendix A, the m^6^A regulators have a varying expression pattern in different immune cell types across the pan-cancer cohort. For instance, many RNA m^6^A regulators have higher expression in resting CD4 memory T cells, resting NK cells, M0 macrophages, M1 macrophages, activated dendritic cells and activated mast cells, but lower expression in memory B cells, plasma cells, naïve CD4 T cells, and T regulatory cells in BLCA. In the case of BRCA, most RNA m^6^A regulators are found to be increased in naïve B cells, resting CD4 memory T cells, M2 macrophages, resting mast cells, and neutrophils, while they are decreased in memory B cells, plasma cells, CD8 T cells, T follicular helper cells, T regulatory cells, and activated NK cells. However, in READ, most RNA m^6^A regulators are found to be decreased in memory B cells and naïve CD4 T cells. Interestingly, most RNA m^6^A regulators are found to be unaffected in UCS (Appendix A).

When we look at the differential immune cell infiltration between clusters in each TCGA pan-cancer cohort, most of the immune cells are found to be affected in BRCA, HNSC, KIRP, KIRC, PRAD, and STAD (Figure 6A). For instance, in the case of the BRCA cancer cohort, the cell percentage of naïve B cells, plasma cells, CD8 T cells, resting CD4 memory T cells, T follicular helper cells, T regulatory cells, gamma delta T cells, activated NK cells, resting dendritic cells, resting mast cells, and neutrophils is found to be significantly different among clusters C1, C2, and C3 (Appendix A). Furthermore, the differential immune cell infiltration analysis between high- and low-risk groups in each TCGA pan-cancer cohort reveals that most of the immune cells are affected in BLCA, KIRC, LGG, LUAD, LIHC, and PAAD (Figure 6B).

### 3.10. Association of Tumor Stemness Score with RNA m^6^A Regulators in TCGA Pan-Cancer Cohort

We investigate the tumor stemness score in the TCGA pan-cancer cohort by analyzing DMPss, DNAss, ENHss, EREG.EXPss, EREG-METHss, and RNAss. The differential tumor stemness score analysis between clusters shows that a few cancer types, including KIRP, KICH, LUAD, LIHC, PAAD, and TGCT are significantly affected (Figure 6A). We also analyze the differential tumor stemness scores between the high- and low-risk groups in each cancer type (Figure 6B). Our results indicate that DMPss and DNAss are positively correlated with the high-risk group in LGG and LUAD. While ENHss is positively correlated with the high-risk group in LGG, it is negatively correlated with the high-risk group in KIRC. Furthermore, EREG.EXPss is positively correlated with the high-risk group in ACC, BLCA, LGG, LUAD, and PAAD, while negatively correlated with the high-risk group in KIRC. EREG-METHss is positively correlated with the high-risk group in LGG, LUAD, and PAAD. RNAss is positively correlated with the high-risk group in ACC, ESCA, LUAD, LIHC, and PAAD, while negatively correlated with the high-risk group in LGG and LAML.

Next, we analyze the correlation between RNA m^6^A modification genes and tumor stemness scores in each cancer type and find both positive and negative correlations between the genes and the stemness scores (Appendix A). For example, the RNA m^6^A regulators are positively correlated with RNAss in ACC, BLCA, BRCA, COAD, ESCA, GBM, HNSC, KIRC, LGG, LUAD, LUSC, LAML, PRAD, PCPG, STAD, TGCT, and UCEC, while they are negatively correlated with RNAss in KIRP and THCA. Additionally, the RNA m^6^A regulators are positively correlated with EREG.EXPss in ACC, KIRC, LGG, LUAD, LAML, PRAD, PAAD, and STAD, whereas they are negatively correlated with EREG.EXPss in BLCA, BRCA, KIRP, LIHC, and THCA. Further, the analysis suggests that several RNA m^6^A regulators are positively correlated with the DMPss, DNAss, ENHss, and EREG-METHss in HNSC, LGG, LUAD, LUSC, STAD, and TGCT, while negatively correlated with BRCA and LIHC. *CBLL1*, *HNRNPA2B1*, *HNRNPC*, *LRPPRC*, *METTL5*, *PRRC2A*, *RBFOX2*, *RBM15*, *TRMT112*, *WTAP*, and *YTHDF2* are usually positively related to EREG.EXPss in ACC, KIRC, LGG, and PRAD; they are also positively related to RNAss in BLCA, LUAD, LUSC, PRAD, PCPG, and STAD.

Some m^6^A regulators are preferred to positively related to most of the stemness factors such as *YTHDF2* in CHOL, *RBM15* in ESCA, *YTHDC2* in GBM; *IGF2BP1*, *IGF2BP2*, *IGF2BP3*, *LRPPRC*, and *YTHDF1* in HNSC; *RBM15B* and *ZC3H13* in KIRC, *ALKBH5*, *HNRNPA2B1*, *IGF2BP1*, *IGF2BP2*, *IGF2BP3*, *RBM15,* and *YTHDF2* in LGG and LUSC; *HNRNPA2B1* and *TRMT112* in PRAD and PCPG; *CBLL1*, *FXR1*, *LRPPRC*, *METTL16*, *METTL5*, *RBM15*, *VIRMA*, *YTHDF2*, and *YTHDF3* in TGCT; and *METTL5* in UCEC (Appendix A).

Some m^6^A regulators are preferred to negatively related to most of the stemness factors such as *METTL14*, *RBM15*, and *WTAP* in BLCA; *METTL14*, *RBM33*, *YTHDC1*, *YTHDC2*, *ZC3H13*, and *ZCCHC4* in BRCA; *ALKBH5*, *FTO*, and *RBFOX2* in COAD; *YTHDC2* in LUAD; *CBLL1*, *HNRNPC*, *IGF2BP2*, *METTL16*, and *YTHDC1* in LIHC; *FTO* in LAML; *METTL14* and *YTHDC2* in PAAD; *RBFOX2* in PCPG; *FTO* and *RBFOX2* in STAD; and *FMR1*, *FTO*, *IGF2BP2*, *IGF2BP3*, *METTL14*, *RBFOX2*, and *RBM15B* in TGCT (Appendix A).

### 3.11. Association of Genomic Heterogeneity with RNA m^6^A Regulators in TCGA Pan-Cancer Cohort

We investigate the genomic heterogeneity score in TCGA pan-cancer cohort by analyzing purity, tumor mutation burden (TMB), homologous recombination deficiency (HRD), heterozygosity (LOH), mutant-allele tumor heterogeneity (MATH), microsatellite instability (MSI), neoantigen (NEO), and ploidy. Our results show differential genomic heterogeneity scores between clusters in each cancer type. Some of the affected cancer types include BRCA, HNSC, KIRC, KICH, LUAD, LUSC, and LIHC, and TBM in LUSC has the highest genomic heterogeneity score among them (Figure 6A). Further, we analyze the differential genomic heterogeneity scores between the high- and low-risk groups in each tumor type (Figure 6B). Our results suggest that purity is negatively correlated with the high-risk group in BLCA, while positively correlated with the high-risk group in KIRP. TMB is positively correlated with the high-risk group in LUAD. HRD is positively correlated with the high-risk group in ACC, BLCA, KIRP, LGG, LUAD, LIHC, and PAAD. LOH is negatively correlated with the high-risk group in ACC, while positively correlated with the high-risk group in BLCA, KIRP, KIRC, LGG, LUAD, LIHC, and PAAD. MATH is positively correlated with the high-risk group in LIHC. NEO is positively correlated with the high-risk group in LUAD. Ploidy is positively correlated with the high-risk group in LIHC.

Further, we analyze the correlation between RNA m^6^A modification genes and genomic heterogeneity scores in each cancer type, and identify both positive and negative correlations between the genes and the genomic heterogeneity scores across different cancer types (Appendix A). For example, many RNA m^6^A regulators are found to be positively correlated with purity in ACC, BLCA, BRCA, CESC, GBM, HNSC, KIRP, LGG, LUAD, LUSC, SKCM, TGCT, and UCEC, while negatively correlated to it in READ. A positive correlation is observed between several RNA m^6^A regulators and TMB in LUAD and READ, whereas negative correlation is observed in KIRP and THCA. While HDR is positively correlated with many RNA m^6^A regulators in ACC, BLCA, GBM, HNSC, KIRP, KICH, LGG, LUAD, LUSC, LIHC, and PRAD, and it is negatively correlated with the regulators in TGCT. Many RNA m^6^A regulators are positively correlated with LOH in BLCA, COAD, HNSC, KIRP, LGG, LUAD, LUSC, and LIHC, whereas they are negatively correlated in KIRC and THCA. There is a positive correlation between RNA m^6^A regulators and MATH in BLCA, BRCA, COAD, LUAD, and LUSC. The MSI is positively correlated with RNA m^6^A regulators in KIRC and LUSC; however, it is negatively correlated with the regulators in PRAD. Lastly, a positive correlation is observed between RNA m^6^A regulators and ploidy in BRCA, COAD, HNSC, LUAD, and LIHC, while a negative correlation is observed in KIRC.

The positive correlation between some genes and some kind of genomic heterogeneity types in some cancer types is more significant than others; for example, the relationships of *IGF2BP1*, *IGF2BP3*, and HRD in ACC; *METTL3* and purity in BLCA; *IGF2BP2* and HRD, LOH in BRCA; *METTL16* and TMB in CESC; *ALKBH5* and TMB, MSI, and NEO in COAD; *YTHDF1* and HRD, LOH, MATH, and ploidy in COAD; *IGF2BP3* and LOH in CHOL; *CBLL1*, *FMR1*, *HNRNPA2B1*, *HNRNPC*, *METTL3*, *PRRC2A*, *RBMX*, and *YTHDC1,* and purity in GBM; *RBFOX2* and NEO in GBM; *METTL5*, and *RBMX*, and purity in HNSC; *FXR1*, *IGF2BP1*, *LRPPRC*, and *METTL5*, and HRD and LOH in HNSC; *FXR1* and HRD in KIRP; *IGF2BP3* and LOH in KIRP; *FXR1*, *IGF2BP1*, and *IGF2BP2*, and MSI and NEO in KIRC; *IGF2BP1*, *TRMT112*, and *ZCCHC4*, and HRD in KICH; *CBLL1*, *METTL16*, and *RBMX*, and purity in LGG; *RBM15*, and *YTHDF2*, and HRD in LGG; *RBMX* and purity in LUAD; *IGF2BP3* and HRD and LOH in LUAD; *HNRNPC*, and *RBMX*, and purity in LUSC; *YTHDF1* and HRD and LOH in LIHC; *HNRNPA2B1*, *METTL3*, and *TRMT112,* and purity, HRD, and LOH in PRAD; *TRMT112* and LOH in PAAD; *ALKBH5*, and *RBM15*, and TMB in READ; *CBLL1*, and *METTL3*, and purity in SKCM; *CBLL1,* and HRD and ploidy in THCA; *IGF2BP2*, *IGF2BP3*, *METTL14*, *RBFOX2*, *RBM15B*, and *RBMX*, and purity in TGCT; and *IGF2BP2* and HRD in UCEC (Appendix A).

The negative correlation between some genes and some kinds of genomic heterogeneity types in some cancer types is more significant than others, such as *METTL5* and LOH in ACC; *IGF2BP2* and purity in BLCA; *METTL14*, *METTL16*, *METTL3*, *YTHDC2*, and *ZCCHC4,* and HRD and LOH in BRCA; *YTHDF1* and TMB, MSI, and NEO in COAD; *PRRC2A*, *RBFOX2*, *RBMX*, *WTAP*, *YTHDC1*, and *YTHDF2*, and LOH in KIRC; *IGF2BP3* and MATH in LGG; *WTAP* and purity in LIHC; *METTL14*, and *YTHDF3,* and MSI in PRAD; *HNRNPC*, and *RBFOX2* and LOH in THCA; and *ALKBH5* and HRD in TGCT (Appendix A).

### 3.12. Association of Immune Regulatory or Checkpoint Genes with RNA m^6^A Regulators in TCGA Pan-Cancer Cohort

We analyze the immune regulatory genes, including chemokines, immunoinhibitors, immunostimulators, MHCs, and receptors, among different clusters in each tumor type and between high- and low-risk groups. We also investigate the immune checkpoint genes in the TCGA pan-cancer cohort by analyzing the inhibitory and stimulatory genes.

Our results demonstrate that the chemokines affect various cancer types among different clusters, including BLCA, BRCA, ESCA, HNSC, KIRP, KICH, LUAD, LUSC, LIHC, PRAD, STAD, SKCM, THCA, TGCT, and UCEC; some chemokines affect some cancer types quite significantly, such as *CX3CL1*, *CXCL9*, *CXCL12*, *CXCL13* and *CXCL16* in HNSC; *CXCL12* in KIRC; *CXCL8* in LUSC, *CXCL8*, *CXCL9*, *CXCL10*, *CXCL11*, *CXCL12, and CXCL16* in PRAD; and *CXCL12* in THCA (Figure 7A). The effect of chemokines is also observed between high- and low-risk groups in each cancer type. While some chemokines are positively correlated, others are negatively correlated with the high-risk group in different cancer types; for example, *CXCL13*, *XCL1*, and *XCL2* in KIRC and *CCL5*, *CXCL9*, *CXCL10*, *CXCL11,* and *CXCL16* in LGG are elevated, while *CCL15* in BLCA, *CX3CL1* and *CXCL12* in KIRC, *CXCL17* in LUAD, *CCL14* and *CCL16* in LIHC, and *CCL14* in PAAD are reduced in the high-risk group (Figure 7B).

The immunoinhibitors affect some cancer types among different clusters, including BLCA, BRCA, CESC, ESCA, GBM, HNSC, KIRP, KIRC, LUAD, LUSC, LIHC, OV, PRAD, PCPG, READ, STAD, THCA, and UCEC; *IL10RB*, *KDR* and *TGFBR1* are significantly affected most tumor types (Figure 7A). Furthermore, a few immunoinhibitors are positively correlated, while some are negatively correlated with the high-risk group in various cancer types; *IL10RB*, *PVRL2*, and *TGFBR1* in most of the tumor types are increased while *ADORA2A* in KIRC and PAAD, and *KDR* and *TGFBR1* in KIRC are the most significantly decreased in the high-risk group (Figure 7B).

The immunostimulators affect various cancer types among different clusters, including BLCA, BRCA, CESC, ESCA, GBM, HNSC, KIRP, KIRC, KICH, LGG, LUAD, LUSC, LIHC, OV, PRAD, READ, STAD, THCA, TGCT, and UCEC; *CD28*, *ENTPD1*, *IL6R*, *NT5E*, *PVR*, *TMEM173*, *TNFSF15*, *TNFSF18* and *TNFSF4* are affected in most tumor types (Figure 7A). Some of the immunostimulators are positively correlated, while a few are negatively correlated with the high-risk group in various cancer types; *CD276*, *MICB*, *PVR*, *TNFSF4*, and *ULBP1* in most tumor types are elevated, while *C10orf54*, *CXCL12*, *ENTPD1*, *IL6R*, *NT5E*, *RAET1E*, *TNFSF13*, and *TNFSF15* in KIRC, and *IL6R*, *TMEM173*, *TNFRSF14*, and *TNFSF15* in LUAD are the most significantly reduced in the high-risk group (Figure 7B).

The MHCs affect cancer types among different clusters, including BLCA, BRCA, CESC, HNSC, KIRP, KIRC, LIHC, PRAD, STAD, and THCA; *B2M*, *HLA-DOA*, *TAP1*, *TAP2,* and *TAPBP* are affected in most tumor types (Figure 7A). As shown in Figure 7B, a few MHCs are positively correlated, while some are negatively correlated with the high-risk group in various cancer types; most of the MHCs in BLCA, LGG, LIHC, and LAML are increased; however, in KIRC and LUAD, they are decreased in the high-risk group.

The receptors affect a few cancer types among different clusters, including BLCA, BRCA, CESC, HNSC, KIRP, KIRC, LUAD, LUSC, LIHC, OV, PRAD, PAAD, THCA, TGCT, and UCEC; *CCR1*, *CCR2*, *CCR4*, *CCR5*, *CCR6*, *CCR8*, *CXCR5*, *XCR1* and *CX3CR1* are affected in most tumor types (Figure 7A). Furthermore, some receptors are positively correlated, while a few are negatively correlated with the high-risk group in various kinds of tumors; most of the receptors in LGG and LIHC are elevated while those in KIRC, LUAD, and PAAD are reduced, especially *CCR6* in LUAD, *CX3CR1* in KIRC, and LUAD in the high-risk group (Figure 7B).

The immune checkpoint inhibitory genes are found to affect various cancer types among different clusters, including BLCA, BRCA, CESC, ESCA, GBM, HNSC, KIRP, KIRC, KICH, LUAD, LUSC, LIHC, OV, PRAD, PCPG, STAD, SKCM, THCA, and UCEC; *CD274*, *CD276*, *EDNRB*, *VEGFA*, and *VEGFB* are the most affected tumor types (Figure 7C). Furthermore, the immune checkpoint inhibitory genes influence the high- and low-risk groups in each cancer type. While some immune checkpoint inhibitory genes are positively correlated, some are negatively correlated with the high-risk group in various cancer types; most immune checkpoint inhibitory genes in LGG and LIHC are increased while those in KIRC and PAAD are decreased, especially *VEGFA* and *C10orf54* in KIRC in the high-risk group (Figure 7D).

The immune checkpoint stimulatory genes affect various TCGA cancer types among different clusters, including BLCA, BRCA, CESC, ESCA, GBM, HNSC, KIRP, KIRC, LGG, LUAD, LUSC, LIHC, OV, PRAD, PAAD, PCPG, READ, STAD, SKCM, THCA, TGCT, and UCEC; *BTN3A1*, *BTN3A2*, *CD28*, *ENTPD1*, *HMGB1*, and *TLR4* are the most affected tumor types (Figure 7C). We also detect the influence of immune checkpoint stimulatory genes between high- and low-risk groups in each cancer type. Some immune checkpoint stimulatory genes are positively correlated, while some are negatively correlated with the high-risk group in various cancer types; most immune checkpoint stimulatory genes in LGG, LIHC, and BLCA are elevated; however, *BTN3A1*, *CX3CL1*, *ENTPD1*, *HMGB1*, and *TLR4* in KIRC; *CD40LG*, *IL2*, and *TNFRSF14* in LUAD; and *CD40LG* and *SELP* in PAAD are significantly reduced in the high-risk group (Figure 7D).

### 3.13. Correlation between microRNAs (miRNAs) and the RNA m^6^A Regulators in TCGA Pan-Cancer

As shown in Appendix A, we analyze the correlations between the miRNAs and RNA m^6^A regulators in the TCGA pan-cancer cohort. Our results demonstrate that most of the genes are positively correlated with miRNAs in CESC, HNSC, LUAD, LIHC, TGCT, and UCEC. However, in the case of LAML, THCA, and UCS, most of the regulators show a negative correlation with miRNAs. We present the top 10 miRNAs, which have the most significant correlation with m^6^A regulators in each cancer type.

### 3.14. Drug Prediction Based on m^6^A Modification Genes in TCGA Pan-Cancer Cohort

To investigate the drug sensitivity in pan-cancer patients, we use *oncoPredict* to predict the relationships between RNA m^6^A regulators and existing drugs in GDSC drug databases. The top 11 predicted drugs are vinorelbine, vinblastine, staurosporine, sepantronium.bromide, paclitaxel, eg5, docetaxel, dinaciclib, daporinad, dactinomycin, and bortezomib. Our results show that most of the genes are negatively correlated with these drugs (based on IC50 values) between high- and low-risk groups in each tumor except for *RBFOX2* in some tumors, *YTHDF1* in SKCM and LUAD, *FTO* in LUSC and TGCT, and *FMR1* in OV and UCS (Appendix A).

### 3.15. Identified m^6^A-Related Biomarkers by Three Machine Learning Algorithms

To investigate whether m^6^A-related genes could serve as a biomarker for prognosis in each cancer type, we use LASSO, SVM-RFE, and RF algorithms to identify the important prognosis-related genes. The biomarkers are identified in 10 cancer types based on the RNA expression (Figure 8A). *IGF2BP1* is found to be a biomarker for BLCA, KICH, KIRP, and UCS; while *IGF2BP2* is found to be a biomarker for ACC, CESC, LAML, LGG, and PAAD; *HNRNPC* is a biomarker for LGG; and *RBMX* is the biomarker for ESCA (Figure 8A). We also identify prognostic biomarkers in each cancer type based on the CNV data with three machine-learning algorithms (Appendix A and Figure 8B). *IGF2BP3* and *YTHDF3* are found to be the biomarkers for BLCA; *METTL14* for CHOL; *RBFOX2* for GBM and KIRP; *RBM15B*, *TRMT112,* and *YTHDF2* for KIRP; *FMR1* and *METTL3* for LGG; *YTHDC2* and *ZCCHC4* for LUAD; *ALKBH5*, *FMR1*, *HNRNPA2B1*, *YTHDC2*, *YTHDF2,* and *ZC3H13* for OV; and *YTHDF2* for PCPG (Figure 8B).

## 4. Discussion

In the current study, we collect a total of 31 m^6^A modification regulators, including 12 writers, 2 erasers, and 17 readers. The mRNA expression, and the genetic and epigenetic alterations of the modifiers are explored by retrieving the expression and genomic data in pan-cancer from the TCGA database. The m^6^A modification regulators are found to be highly expressed across various cancers and most of them harbored genetic aberrations, including SNVs and CNVs. Furthermore, the differential expression analysis between the cancer and normal tissues of the m^6^A modifiers reveals significant differences for most regulators. Additionally, the m^6^A regulators are found to be significantly correlated with the clinical parameters, such as age, gender, tumor stage, and grade in pan-cancer. Based on the expression of m^6^A modifiers, we classify the tumors into different clusters or high- and low-risk groups. The m^6^A modifiers are then evaluated among different clusters or between the high- and low-risk groups in the context of tumor microenvironment infiltrating immune cells, various tumor stemness, genomic heterogeneity properties, and immune regulatory or checkpoint genes. Lastly, the m^6^A modifiers are correlated with miRNA expression and drug sensitivity in different pan-cancer clusters as well as high- and low-risk tumor groups. Finally, we identify some m^6^A-related biomarkers using three machine-learning algorithms: LASSO, support vector machine-recursive feature elimination (SVM-RFE), and random forest (RF), according to their RNA expression or CNV data. For example, *IGF2BP1*, *IGF2BP2*, *HNRNPC,* and *RBMX* are found to be prognostic biomarkers based on their RNA levels. *IGF2BP3*, *YTHDF3*, *METTL14*, *RBFOX2*, *RBM15B*, *TRMT112*, *YTHDF2*, *FMR1*, *METTL3*, *YTHDC2*, *ZCCHC4*, *ALKBH5*, *HNRNPA2B1*, and *ZC3H13* are found to be prognostic biomarkers based on their CNV levels.

m^6^A RNA modification is associated with multiple processes related to cancer, such as tumorigenesis, drug resistance, tumor epithelial-mesenchymal transition (EMT), and tumor metastasis. It also contributes to the self-renewal and differentiation of cancer stem cells, and resistance to radiotherapy and chemotherapy [64]. Some evidence agrees with the high/low expression of m^6^A regulators across different cancers in our pan-cancer analysis. Accumulating evidence suggests that m^6^A regulators act both as promoters and suppressors of oncogenesis. This is achieved by directly promoting/inhibiting the expression of m^6^A regulators or indirectly influencing the downstream oncogenes or tumor suppressors during different stages or in different cancer types [65]. For instance, *METTL3* is highly expressed and promotes bladder cancer by promoting the expressions of *MYC* and *AFF4*; inhibition of *METTL3* reduces bladder tumor cell proliferation, migration, and invasion [66]. In endometrial tumors, reduced *METTL3* expression or *METTL14* mutation causes reduced levels of m^6^A, leading to enhanced cell proliferation, colony formation, migration, and invasion of tumor cells [65,67].

When the genomic alterations are investigated, most of the m^6^A regulators harbor SNVs across different cancers, with *ZC3H13*, *VIRMA*, and *PRRC2A* having higher mutation frequencies. Multiple studies have reported genomic variations in m^6^A regulators in different cancers [68,69,70]. *ZC3H13* (zinc finger CCCH-type containing 13) is a canonical zinc finger protein, which harbors a somatic frame-shift mutation in colorectal cancer [68,71]. *VIRMA* (vir-like m^6^A methyltransferase associated) is known to promote the progression of cancer and is associated with poor survival in multiple types of cancer [72]. *PRRC2A* (Proline-rich coiled-coil2A), also named *BAT2*, is localized near the genes coding for TNF alpha and TNF beta, and all these genes are within the human major histocompatibility complex class III region [73]. Furthermore, mutations in *PRRC2A* have been found to be associated with multiple cancers [74,75,76].

The survival analysis suggests involvement of the m^6^A regulators with overall survival, both positively and negatively. For instance, *METTL3* is found to be positively associated with the prognosis of LIHC and LAML; it is negatively associated with PAAD. Similarly, another gene from the same family, *METTL14*, is positively associated with the prognosis of LGG and LAML, while negatively associated with KIRC, SKCM, and READ. There is accumulating biological evidence that these genes are associated with the prognosis of multiple cancers. A meta-analysis by Liu et al. [77] suggests that the upregulation of *METTL3* is significantly associated with poor prognosis. *METTL14* (Methyltransferase-like 14) is the central component of the m^6^A methyltransferase complex and acts as both an oncogene and tumor suppressor gene. A review by Guan et al. [78] systematically summarizes the latest research on *METTL14*. The downregulated *METTL14* acts as a tumor suppressor in breast cancer and predicts poor prognosis [79].

Furthermore, we find a significant association between the clinicopathological parameters and the m^6^A modification regulators in the current study. Such an association has been reported by multiple studies in literature. For instance, a meta-analysis by Su et al. [80] on human cancers identifies *METTL3* and *METTL14* to be the most important prognostic markers in cancer. A study by Zheng et al. [81] evaluates the relationship between m^6^A modification and clinicopathological characteristics in breast cancer and identifies *CBLL1* as a promising prognostic biomarker.

Further, we classify the cancer patients based on the similarity of their m^6^A regulators’ levels into two to four clusters using the consensus-cluster method and evaluate the differential expression of the m^6^A regulators across different clusters. The consensus-cluster method has been widely used to analyze m^6^A regulators-related immune characteristics in various conditions, including spondylitis [82], prostate cancer [83], AML [84], and hepatitis B virus-related hepatocellular carcinoma [85].

The LASSO Cox regression method is suitable for constructing models when there are a large number of independent variables and in the case of restricted sample size [86]. Previously, LASSO regression has been used to construct the m^6^A regulators-based risk signature in various conditions, including head and neck cancer [87], colorectal cancer [88,89], hepatocellular carcinoma [90], and AML [84]. We apply the LASSO Cox regression model to the pan-cancer cohort to identify high- and low-risk groups based on m^6^A regulators-based prognostic signatures. Different sets of prognostic signatures are identified for different cancers by us, and *IGF2BP3*, a well-known m^6^A regulator, is found to be highly expressed in a high-risk group of several cancers. The importance of this gene in cancer prognosis has been shown by other studies as well [91,92,93].

We further evaluate the differential immune cell infiltration between the high- and low-risk groups. Tumor–immune cell infiltration is closely related to clinical outcomes and the composition of tumor-infiltrating immune cells can serve as a diagnostic and prognostic biomarker [94,95,96]. Our results indicate a significant differential infiltration rate of multiple immune cell types between high- and low-risk KIRC along with a few other cancer types including LUAD, LGG, BLCA, and LIHC. A study by Zuo et al. [96] on immune cell infiltration patterns across 32 cancer types reveals that patients with high immune cell infiltration have worse OS, but PFS compared to those with low immune cell infiltration. Further, the authors suggest considerable heterogeneity in the prognostic value of these cells in different cancer types, which is in agreement with our results on pan-cancer analysis. Another pan-cancer study by Guo et al. [97] suggests that *METTL3* regulates the tumor immune microenvironment and epithelial–mesenchymal transition by modulating RNA modification and metabolism.

The correlation of the m^6^A regulators with immune checkpoint genes and immune regulatory genes, including chemokines, immunoinhibitors, immunostimulators, and MHCs, results in significant correlations across different cancers.

We investigate the tumor stemness score in the TCGA pan-cancer cohort by analyzing DMPss, DNAss, ENHss, EREG.EXPss, EREG-METHss, and RNAss. The tumor stemness score of a few cancer types, including KIRP, KICH, LUAD, LIHC, PAAD, and TGCT, are significantly varied among different clusters. Also, there is an obvious discrepancy in tumor stemness between high- and low-risk groups in multiple cancer types.

Additionally, we investigate the genomic heterogeneity score in pan-cancer by analyzing different parameters, such as TMB, HRD, LOH, MSI, MATH, and NEO. Our analysis show significant differences in different parameters between the high- and low-risk groups across different tumor types. When the m^6^A modification genes are correlated with the heterogeneity scores, we observe significant positive and negative correlations in different cancer types. The genomic heterogeneity properties have been shown to be useful as prognostic biomarkers and in therapy selection [98]. Recent studies have shed light on the key roles of m^6^A modifiers in modulating DNA repair and genome integrity and stability [99]. A few m^6^A regulators can regulate the RNA levels involved in DNA damage and repair, in turn affecting the genomic instability [100].

Next, we analyze the correlations between m^6^A regulators and miRNAs in pan-cancer. Overall, our results show a positive correlation between the regulators and miRNAs in several cancer types. Recently, increasing evidence has suggested an interplay between miRNAs and m^6^A modification. m^6^A modification plays an important role in regulating miRNA biosynthesis, while miRNAs affect m^6^A levels by targeting m^6^A regulatory RNAs. A review by Han et al. [101] discusses, in detail, the interaction between m^6^A modification and miRNAs.

Further, the drug sensitivity analysis predicts the relationship between the m^6^A regulators and multiple drugs. Some of the top drugs that are identified to target these genes include vinorelbine, vinblastine, and staurosporine. Vinorelbine, an alkaloid, is an antineoplastic drug and is used as first-line chemotherapy for metastatic cancers [102]. Vinblastine, another alkaloid, is used to treat mainly blood cancers by inducing acute cell cycle phase-independent apoptosis [103,104]. Staurosporine, a highly successful anti-cancer drug, is produced by a soil-dwelling microbe. It acts by inhibiting protein kinases, particularly tyrosine kinases, and has a remarkably strong cytotoxic effect on cancer cells [105].

Although our study obtain several interesting findings, there are a few limitations. In this study, several important molecules are found to affect different cancers; however, further validations using other datasets as well as experiments are required to confirm their exact roles. Furthermore, additional external validations using other cohorts are needed to evaluate whether the m^6^A molecular subtypes and risk-groups still perform well in various cancer types.

## Figures and Tables

**Figure 1 biomedicines-12-02211-f001:**
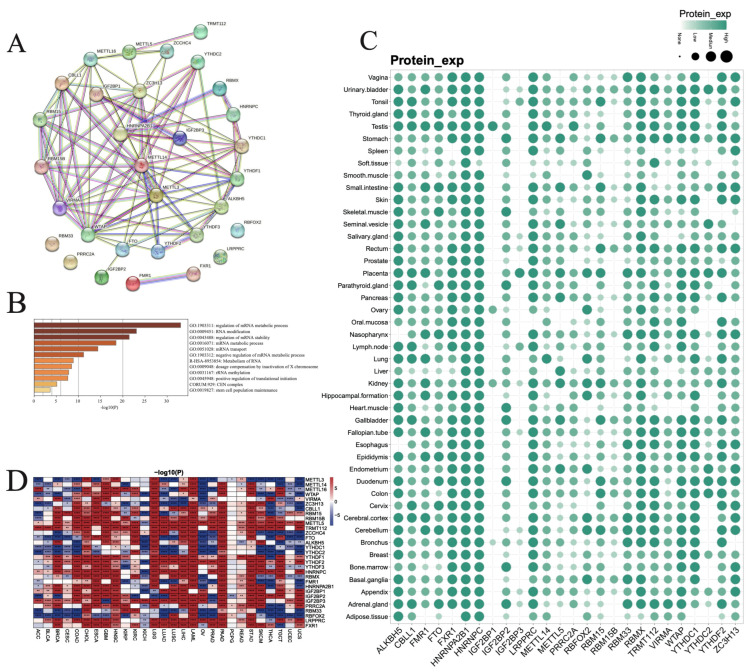
Expression levels of m^6^A modification genes across cancer types. (**A**) Protein–protein interaction (PPI) analysis of genes. (**B**) GO annotation of genes. (**C**) Protein expression levels of genes in various tissues. (**D**) Expression differences in various normal and cancer tissues. (* *p* < 0.05, ** *p* < 0.01, *** *p* < 0.001, **** *p* < 0.0001).

**Figure 2 biomedicines-12-02211-f002:**
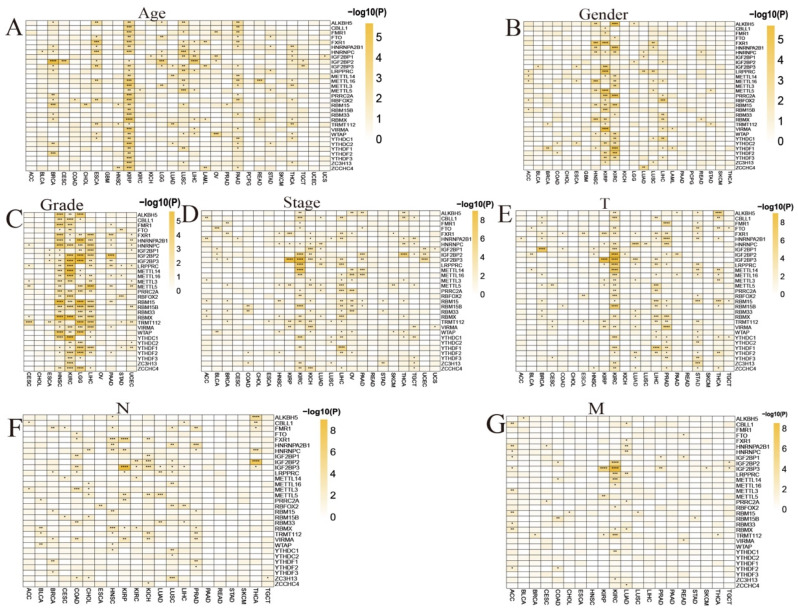
Clinicopathological characteristics of m^6^A modification genes in each cancer. Expression of m^6^A modification genes between (**A**) age groups, (**B**) gender, (**C**) grade, (**D**) stage, (**E**) T, (**F**) N, and (**G**) M groups. (* *p* < 0.05, ** *p* < 0.01, *** *p* < 0.001, **** *p* < 0.0001).

**Figure 3 biomedicines-12-02211-f003:**
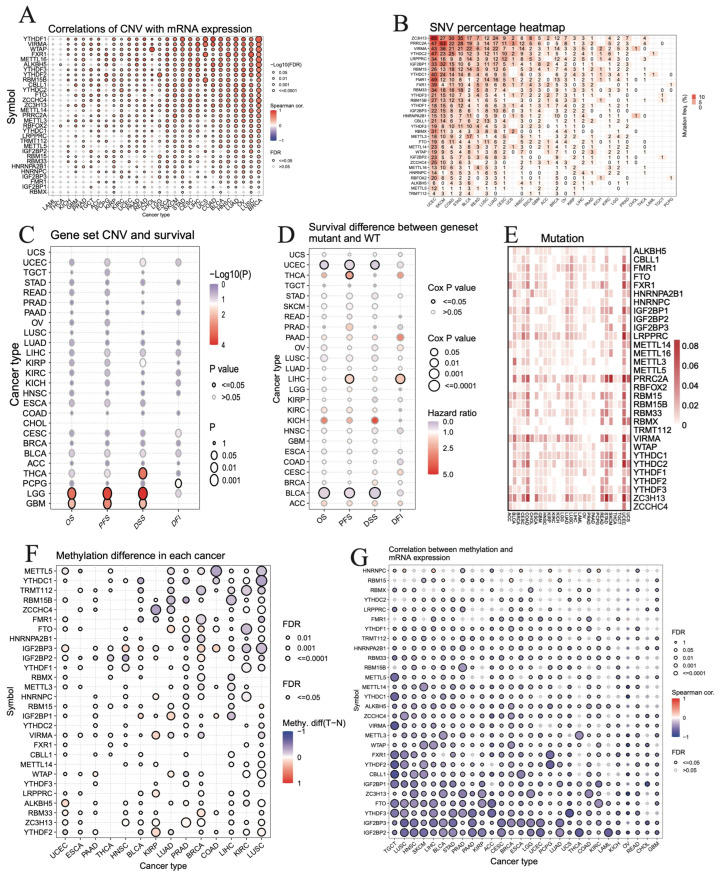
Genetic and epigenetic alterations of m^6^A modification genes in each cancer. (**A**) Correlations between the CNV and RNA expressions of genes in each cancer. (**B**) Heatmap showing the SNV percentage in the genes in each cancer. (**C**) Association between geneset CNV and survival status in each cancer. (**D**) Association between geneset SNV mutation and survival status in each cancer. (**E**) Heatmap showing the genomic variations, including missense mutation, nonsense mutation, frame-shift deletion, frame-shift insertion, in-frame deletion, and in-frame insertion of genes in different cancer types. (**F**) Methylation difference in each gene in each cancer. (**G**) Correlation between the methylation level and RNA expression of genes in each cancer.

**Figure 4 biomedicines-12-02211-f004:**
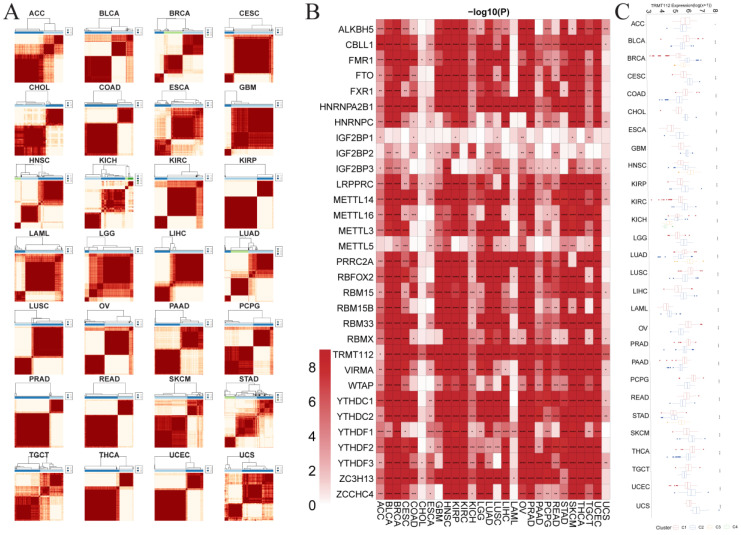
Consensus cluster analysis of m^6^A modification genes in each cancer. (**A**) Consensus cluster analysis of different tumors. (**B**) Heatmap showing the differential expression of genes between the clusters in each tumor. (**C**) Differential expression of *TRMT112* between the clusters in each tumor. (* *p* < 0.05, ** *p* < 0.01, *** *p* < 0.001, **** *p* < 0.0001).

**Figure 5 biomedicines-12-02211-f005:**
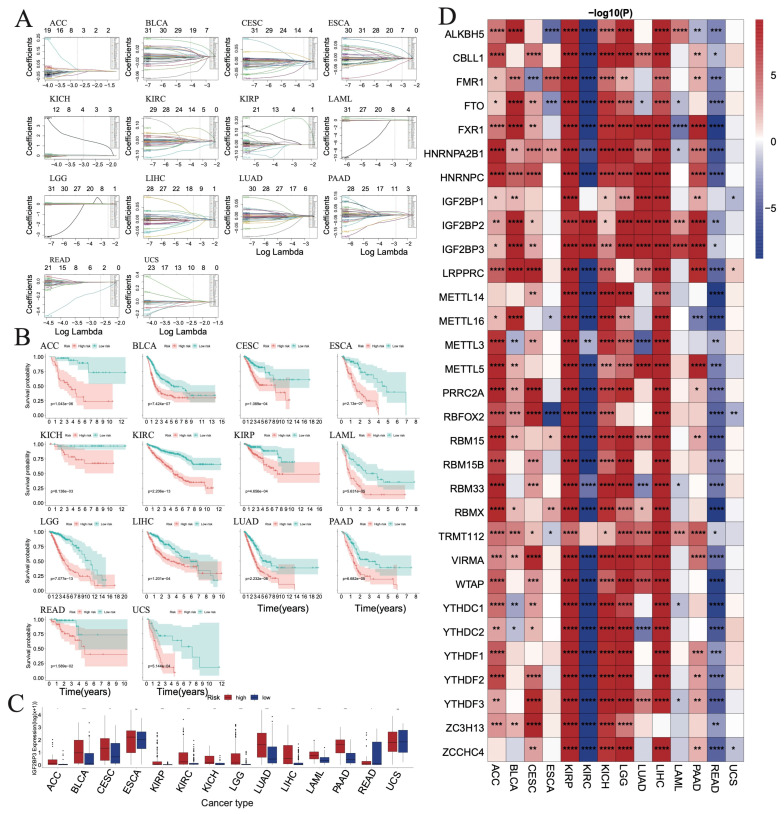
Prognostic model of m^6^A modification genes in each cancer. (**A**) LASSO regression is used to screen for important genes in each tumor. (**B**) Survival curves of patients between high- and low-risk groups in each tumor. (**C**) Differential expression of *IGF2BP3* between high- and low-risk groups in each tumor. (**D**) Heatmap showing the differential expression of genes between high- and low-risk groups in each tumor. (* *p* < 0.05, ** *p* < 0.01, *** *p* < 0.001, **** *p* < 0.0001).

**Figure 6 biomedicines-12-02211-f006:**
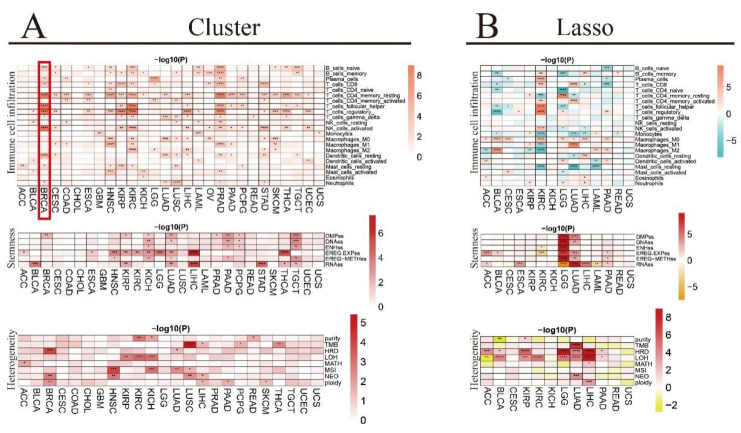
Association of immune cell infiltration, tumor stemness score, and genomic heterogeneity with RNA m^6^A regulators in TCGA pan-cancer cohort. (**A**) Differential immune cell infiltration, tumor stemness score, and genomic heterogeneity between clusters in each tumor. (**B**) Differential immune cell infiltration, tumor stemness score, and genomic heterogeneity between high- and low-risk groups in each tumor. (* *p* < 0.05, ** *p* < 0.01, *** *p* < 0.001, **** *p* < 0.0001).

**Figure 7 biomedicines-12-02211-f007:**
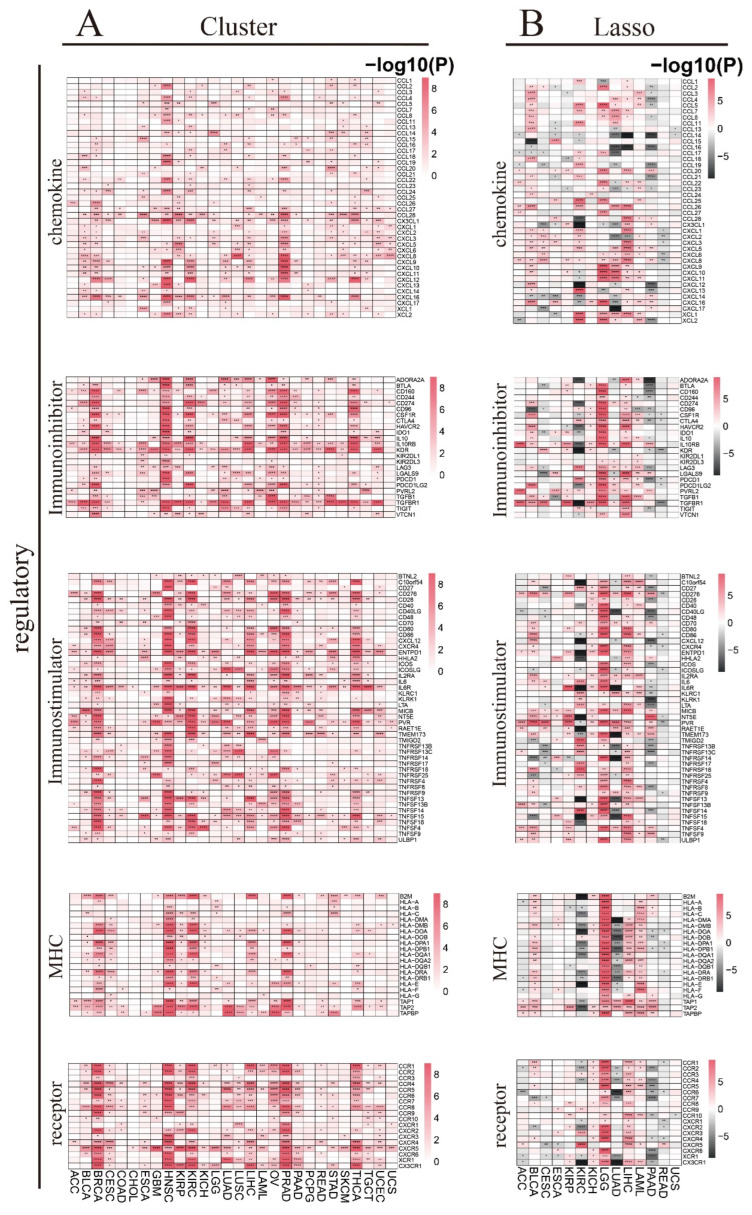
Association of immune regulatory and checkpoint genes with RNA m^6^A regulators in multiple cancers. (**A**) Differential immune regulatory genes including chemokine, immunoinhibitor, immunostimulator, MHC, and receptor genes between clusters in each cancer type. (**B**) Differential immune regulatory genes including chemokine, immunoinhibitor, immunostimulator, MHC, and receptor genes between high- and low-risk groups in each cancer type. (**C**) Differential immune checkpoint inhibitory and stimulatory genes between clusters in each cancer type. (**D**) Differential immune checkpoint inhibitory and stimulatory genes between high- and low-risk groups in each cancer type. (* *p* < 0.05, ** *p* < 0.01, *** *p* < 0.001, **** *p* < 0.0001).

**Figure 8 biomedicines-12-02211-f008:**
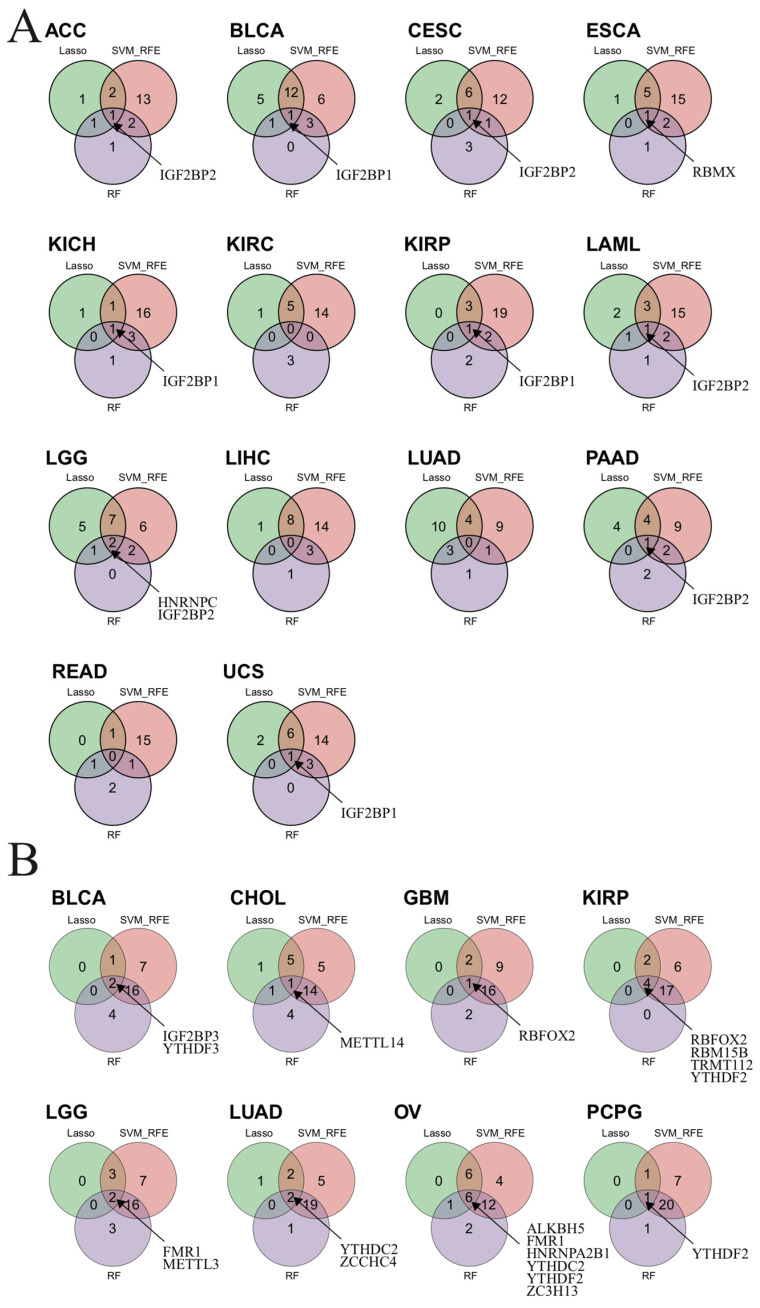
Identifying m^6^A-related biomarkers using three machine-learning algorithms. (**A**) The Venn plot shows biomarkers obtained from the intersection of results from SVM-RFE, RF, and LASSO algorithms, based on the RNA expression levels in each cancer type. (**B**) The Venn plot shows biomarkers obtained from the intersection of results from SVM-RFE, RF, and LASSO algorithms based on the CNV data in each cancer type.

## Data Availability

The datasets presented in this study can be found in online repositories.

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
