# Peer review of "Interplay of RNA m6A Modification-Related Geneset in Pan-Cancer"

_biomedicines, 2024, doi:10.3390/biomedicines12102211_

Round 1

Reviewer 1 Report

Comments and Suggestions for Authors

Review of

Interplay of RNA m6A modification related geneset in pan-cancer

Overall comments

The manuscript presents a lot of data and show significant correlation of genes involved in m6A modifications with various characteristics of cancers.

However, m6A genes investigated in this manuscript (listed in Suppl table 1) have already been shown to be cancer-promoting in various publications which the authors themselves point in the discussion and list in the supplemental Table 2. Thus, it is no surprise that most of them positively correlate with the pan-Cancer TCGA.

The authors use a vast repertoire of bioinformatics tools to analyze a lot of data with a lot of statistics, but in all this they do not show any biological significance and functional context. This is necessary to advance our knowledge of why m6A modifications have a role in cancer.

I would recommend to summarize the data so the reader gets an overall grasp of the data and add biological context to the statistical significance

This manuscript has an exceptional high number of 16 figures. Additionally, most of the graphs in all those figures are so small that to be unreadable. I would recommend to condense the data to a few key readable figures and move the rest to the supplement albeit with increased panel size to be readable.

While chapters 3.1 to 3.11 focus mostly on genes related to m6A modifications chapters 3.12 to 3.16 are unrelated analysis of the TCGA pan-cancer cohort. I think those chapters may better be published separately.

Specific comments:

Abstract:

The abstract is too descriptive on the methods, especially between lines 16 and 28. The authors should rewrite in focusing on the results and their biological significance.

Introduction:

The authors should also give some information and references about the other writers besides METTL3 and YTHDF1, readers, erasers genes. Why is SOCS introduced?

All figures: Please increase the graphs in all figures so the reader can decipher their content. If not possible only show the most important ones and add the rest to the supplemental material.

Suppl. Table 1: Please add references to the main manuscript’s reference list

Fig.1A: This panel does not contribute to the results. It is not relevant where those genes are located on the genome.

Fig16: Please remove the gibberish on top of each panel.

Code availability:

I appreciate the publishing of the code in github. Please translate the r scripts from Chinese into English

Comments on the Quality of English Language

 Minor editing of English language required

Author Response

Overall comments

The manuscript presents a lot of data and show significant correlation of genes involved in m6A modifications with various characteristics of cancers.

However, m6A genes investigated in this manuscript (listed in Suppl table 1) have already been shown to be cancer-promoting in various publications which the authors themselves point in the discussion and list in the supplemental Table 2. Thus, it is no surprise that most of them positively correlate with the pan-Cancer TCGA.

The authors use a vast repertoire of bioinformatics tools to analyze a lot of data with a lot of statistics, but in all this they do not show any biological significance and functional context. This is necessary to advance our knowledge of why m6A modifications have a role in cancer.

I would recommend to summarize the data so the reader gets an overall grasp of the data and add biological context to the statistical significance.

This manuscript has an exceptional high number of 16 figures. Additionally, most of the graphs in all those figures are so small that to be unreadable. I would recommend to condense the data to a few key readable figures and move the rest to the supplement albeit with increased panel size to be readable.

While chapters 3.1 to 3.11 focus mostly on genes related to m6A modifications chapters 3.12 to 3.16 are unrelated analysis of the TCGA pan-cancer cohort. I think those chapters may better be published separately.

Response: Thanks for your comments. Chapters 3.12 to 3.16 are the analysis of the different clusters or high- and low-risk groups in each TCGA cancer cohort. These clusters of high- and low-risk groups are grouped based on the levels of RNA m6A related geneset in patients. Therefore, the chapters 3.12 to 3.16 are relevant to chapters 3.1 to 3.11. Furthermore, in the revised version, we have kept only the key figures that have now been improved for better readability and clarity. The other figures have been moved to the Supplementary.

Specific comments:

Abstract:

The abstract is too descriptive on the methods, especially between lines 16 and 28. The authors should rewrite in focusing on the results and their biological significance.

Response: Thanks for your critical suggestion. This is indeed a very useful point. For the abstract, we have simplified the methods and focused on the results and their biological significance in the revised version.

Introduction:

The authors should also give some information and references about the other writers besides METTL3 and YTHDF1, readers, erasers genes. Why is SOCS introduced?

Response: We apologize for missing the information about other RNA m6A related genes. In the revised version, we have included them briefly. SOCS was introduced by mistake, which has now been removed in the revised version.

All figures: Please increase the graphs in all figures so the reader can decipher their content. If not possible only show the most important ones and add the rest to the supplemental material.

Response: Thanks for your insightful suggestion. We have improved the quality of the graphs in all figures in the revised version and included only the important ones in the main manuscript. The other figures have been moved to the Supplementary.

Suppl. Table 1: Please add references to the main manuscript’s reference list.

Response: Thanks for your useful suggestion. Sorry for missing this part of information in the main text reference list. We have supplemented it in the revised version.

Fig.1A: This panel does not contribute to the results. It is not relevant where those genes are located on the genome.

Response: Thanks for bringing up this point. We have deleted it in the revised version.

Fig16: Please remove the gibberish on top of each panel.

Response: Thanks for the careful checking. We are sorry for the unclear gibberish and have removed it in the revised version.

Code availability:

I appreciate the publishing of the code in github. Please translate the r scripts from Chinese into English.

Response: Thanks for the constructive advice. We are sorry for the Chinese version and have translated it into English in the revised version.

Comments on the Quality of English Language

Minor editing of English language required

Response: Thanks for the suggestion. We have polished it with the help of a English-speaking colleague.

Reviewer 2 Report

Comments and Suggestions for Authors

The objective of the study by Zhang et al. is to analyze the expression patterns, functions and genomic alterations of RNA m6A modification regulators in different cancer types obtained from TCGA. The authors also analyzed the overall survival (OS) status based on the expression of RNA m6A modification regulators and estimated their clinical significance, and developed a LASSO Cox regression model to evaluate the prognostic value of RNA m6A regulators. In addition, the authors studied the relationship of RNA m6A regulators with various tumor features, such as immune cell infiltration, microRNAs, predicted drug sensitivity, tumor stemness score, genomic heterogeneity, immune regulatory and immune checkpoint genes.

Overall, the authors conducted a great number of analyses of RNA m6A regulators, which could provide a valuable resource for scientists interested in this topic. However, the sheer number of analyses makes the study very descriptive and quite difficult to follow. In my opinion the analysis would benefit from reducing the number of figures, and an effort to consolidate certain results. More specific comments below:

11.     Most of the study comprises of analyses that demonstrate varying results for different cancer types, for example “Our results revealed that the m6A regulators were highly expressed in most of the cancer types such as CHOL, ESCA, GBM, HNSC, LGG, LAML, PAAD and STAD, while they were decreased in a few tumor types, including ACC, KICH, OV, THCA and UCS “ but the authors make no effort to further analyze and explain the differences observed in different cancer types. For instance KIRP, KIRC, LGG, LIHC, seem to show consistent results in various analyses (eg. Figure 4, Figure 5, Figure 9), but the authors do not comment on this nor do they try to provide an explanation for the differences in results of different cancer type. Are the differences related to the different number of samples for different cohorts? What is the number of samples used for each cancer type? Do different cancer types also include various subtypes? Is this related to consistency of results?

22.      I don’t really see the added value of analyzing the similarity of TME infiltrating cells (used to define clusters via consensus clustering), since the main objective of the study are RNA m6A modifiers. If such an analysis must be included then all the results related to the characterization of the clusters should be in one section, to minimize the confusion of the reader. (For instance Fig9 B and C should be put together with Figure 7). Are the clusters in different tumors similar?

33.      The quality of the figures is really low and in most cases the letters and numbers cannot be read even with high magnification, so I have no way of assessing the validity of the results described in the text. This should be addressed in the majority of figures (Figure 2, Figure3B, Figure 5, Figure 6, Figure 8ABC, Figure 9A, Figure 11B, Figure 12, Figure 13C, Figure 14C)

44.      Figure 16 includes unrecognized symbols.

55.      In the entire study it’s unclear how the authors choose genes/tumor types for more specific analysis (eg. Figure 9B, Figure 11B)

66.      In Figure 10 the authors show results for about 6-7 miRNAs  for each cancer type. How were the miRNAs chosen for analysis.

77.      The analysis of clinicopathological characteristics of m6A modification genes in each cancer (Figure 5) does not include the same cancer types in each analysis. What is the reason for this. What is the number of samples that have available clinical data for each cancer type and each analysis? The authors should include information about the distribution of age groups, gender, stage, grade, etc. for each cancer cohort in the Supplementary information  and check whether the number of missing data influences their results for each cancer type.

88.      The authors show that several clinicopathological characteristics are related to m6A modification genes. To exclude the influence of those features on the LASSO Cox regression model used to evaluate the prognostic value of RNA m6A regulators, the authors should train a stratified Cox regression model.

99.      Figure 6C shows the same information as Figure 3A and should be excluded from the manuscript.

Comments on the Quality of English Language

The article would benefit from proofreading and minor editing of English. 

Author Response

Comments and Suggestions for Authors

The objective of the study by Zhang et al. is to analyze the expression patterns, functions and genomic alterations of RNA m6A modification regulators in different cancer types obtained from TCGA. The authors also analyzed the overall survival (OS) status based on the expression of RNA m6A modification regulators and estimated their clinical significance, and developed a LASSO Cox regression model to evaluate the prognostic value of RNA m6A regulators. In addition, the authors studied the relationship of RNA m6A regulators with various tumor features, such as immune cell infiltration, microRNAs, predicted drug sensitivity, tumor stemness score, genomic heterogeneity, immune regulatory and immune checkpoint genes.

Overall, the authors conducted a great number of analyses of RNA m6A regulators, which could provide a valuable resource for scientists interested in this topic. However, the sheer number of analyses makes the study very descriptive and quite difficult to follow. In my opinion the analysis would benefit from reducing the number of figures, and an effort to consolidate certain results. More specific comments below:

Response: We thank the reviewer for these comments. We have condensed our figures to a few key ones and moved others to the supplementary figures.

  1. Most of the study comprises of analyses that demonstrate varying results for different cancer types, for example “Our results revealed that the m6A regulators were highly expressed in most of the cancer types such as CHOL, ESCA, GBM, HNSC, LGG, LAML, PAAD and STAD, while they were decreased in a few tumor types, including ACC, KICH, OV, THCA and UCS “ but the authors make no effort to further analyze and explain the differences observed in different cancer types. For instance KIRP, KIRC, LGG, LIHC, seem to show consistent results in various analyses (eg. Figure 4, Figure 5, Figure 9), but the authors do not comment on this nor do they try to provide an explanation for the differences in results of different cancer type. Are the differences related to the different number of samples for different cohorts? What is the number of samples used for each cancer type? Do different cancer types also include various subtypes? Is this related to consistency of results?

Response: We thank the reviewer for bringing up this. This is indeed a very critical point. The varying results for different cancer types are mainly due to the specific pathological features of each cancer type. This is the question of tissue specificity and not the number of samples for different cohorts, because there are enough number of samples in each cohort. We have now added the number of samples used for each cancer type in supplementary table 2. We could not find the subtype information in some cancer types, and in a few cases the subtypes are listed as independent cancer types (e.g., KICH, KIRC and KIRP for kidney cancer; LUAD and LUSC for lung cancer).

  1. I don’t really see the added value of analyzing the similarity of TME infiltrating cells (used to define clusters via consensus clustering), since the main objective of the study are RNA m6A modifiers. If such an analysis must be included then all the results related to the characterization of the clusters should be in one section, to minimize the confusion of the reader. (For instance Fig9 B and C should be put together with Figure 7). Are the clusters in different tumors similar?

Response: We thank the reviewer for this insightful suggestion. We have now reorganized the figures to make them clearer and more readable. We have put Figures 9B and 9C together with Figures 13A, 13B, 14A and 14B to make a new Figures 8A and 8B to illustrate the discrepancy of immune cell infiltration, tumor stemness and genomic heterogeneity in different clusters and high-/low-risk groups. Different tumors have been classified to different clusters based on the levels of RNA m6A related geneset in patients, not the similarity of TME infiltrating cells.

  1. The quality of the figures is really low and in most cases the letters and numbers cannot be read even with high magnification, so I have no way of assessing the validity of the results described in the text. This should be addressed in the majority of figures (Figure 2, Figure3B, Figure 5, Figure 6, Figure 8ABC, Figure 9A, Figure 11B, Figure 12, Figure 13C, Figure 14C)

Response: We apologize for the low quality of the figures. We have now improved the text and numbers in the figures for better readability and clarity in the revised version.

  1. Figure 16 includes unrecognized symbols.

Response: We thank the reviewer for the careful checking. We apologize for this and have now removed the unclear gibberish in the revised version.

  1. In the entire study it’s unclear how the authors choose genes/tumor types for more specific analysis (eg. Figure 9B, Figure 11B)

Response: We apologize for not explaining this point clearly in our previous manuscript. Generally, for the entire study, we have mainly chosen genes/tumor types based on the significant differences in the results.

For example, Figure 9B (new Figure 8A) represents the differential immune cell infiltration among clusters, and we selected BRCA for further presentation, as shown in Figure 9C (new supplementary Figure SF4B) because it has the most significant difference of immune cell infiltration in different clusters among all the tumor types.

Figure 11B (new Figure 7B) shows the correlation coefficient and p-values between m6A related genes and ImmuneScore in cancers. It only shows a representative one with the most significant p-value in each cancer type.

  1. In Figure 10 the authors show results for about 6-7 miRNAs for each cancer type. How were the miRNAs chosen for analysis.

Response: We apologize for not presenting this clearly. The miRNAs chosen by their correlation with m6A related genes. We chose the top 10 miRNAs most significantly correlated with m6A related genes.

  1. The analysis of clinicopathological characteristics of m6A modification genes in each cancer (Figure 5) does not include the same cancer types in each analysis. What is the reason for this. What is the number of samples that have available clinical data for each cancer type and each analysis? The authors should include information about the distribution of age groups, gender, stage, grade, etc. for each cancer cohort in the Supplementary information and check whether the number of missing data influences their results for each cancer type.

Response: Thanks for the critical question. We could not find the clinicopathological data in some cancer types. We have added the number of samples used for each cancer type in supplementary table 2. We also added the distribution of age groups, gender, stage, grade, etc. for each cancer cohort in the supplementary data.

T2

T1

T3

T4

N0

N1

N2

N3

M0

M1

MALE

FEMALE

G3

G2

G1

G4

YOUNG

OLD

Stage II

Stage I

Stage III

Stage IV

ACC

41

9

8

17

66

9

0

0

60

15

31

46

0

0

0

0

57

20

36

9

15

15

BLCA

150

3

195

58

237

46

74

8

196

11

301

106

0

0

0

0

87

320

130

0

140

133

BRCA

630

280

138

40

514

361

119

77

907

22

12

1079

0

0

0

0

580

510

617

182

248

20

CESC

71

140

20

10

133

60

0

0

116

10

0

0

118

135

18

0

239

65

69

162

45

21

COAD

44

6

196

40

166

71

49

0

193

40

156

130

0

0

0

0

92

194

110

44

82

40

CHOL

12

19

5

0

26

5

0

0

28

5

16

20

18

15

0

0

12

24

9

19

0

7

ESCA

43

31

97

6

76

78

13

7

141

16

155

26

49

74

18

0

82

99

80

18

61

16

GBM

0

0

0

0

0

0

0

0

0

0

98

54

0

0

0

0

71

81

0

0

0

0

HNSC

157

51

120

189

220

78

205

11

502

5

382

136

124

304

61

7

232

285

82

27

93

316

KIRP

33

191

60

0

144

25

4

0

206

12

213

75

0

0

0

0

120

165

25

177

52

16

KIRC

69

272

179

10

240

15

0

0

441

79

344

186

206

228

14

74

243

287

57

266

123

81

KICH

25

21

18

0

40

3

0

0

0

0

39

27

0

0

0

0

47

19

25

21

14

6

LGG

0

0

0

0

0

0

0

0

0

0

283

225

260

247

0

0

439

69

0

0

0

0

LUAD

276

169

47

18

329

96

74

0

344

25

237

276

0

0

0

0

137

357

122

274

83

26

LUSC

292

112

71

23

317

130

40

5

408

7

369

129

0

0

0

0

90

399

161

242

84

7

LIHC

94

179

80

13

250

4

0

0

264

4

249

120

121

177

55

11

168

200

86

169

85

5

LAML

0

0

0

0

0

0

0

0

0

0

93

80

0

0

0

0

90

83

0

0

0

0

OV

0

0

0

0

0

0

0

0

0

0

0

0

360

47

0

0

220

199

24

0

328

63

PRAD

188

0

292

10

343

79

0

0

453

3

0

0

0

0

0

0

201

294

0

0

0

0

PAAD

24

7

142

3

49

124

0

0

80

4

98

80

48

95

31

0

55

123

147

21

3

4

PCPG

0

0

0

0

0

0

0

0

0

0

77

100

0

0

0

0

136

41

0

0

0

0

READ

13

4

63

10

38

29

21

0

63

12

49

42

0

0

0

0

37

54

24

12

33

13

STAD

89

21

180

115

124

111

78

82

366

27

268

146

245

148

12

0

122

287

121

58

169

41

SKCM

5

0

10

84

58

8

10

10

97

3

60

42

0

0

0

0

37

65

66

0

26

3

THCA

166

142

171

23

229

225

0

0

281

9

136

368

0

0

0

0

384

120

52

283

112

55

TGCT

50

76

6

0

79

22

3

4

126

6

0

0

0

0

0

0

130

2

13

104

14

0

UCEC

0

0

0

0

0

0

0

0

0

0

0

0

141

21

14

0

48

129

24

98

48

10

UCS

0

0

0

0

0

0

0

0

0

0

0

0

0

0

0

0

6

51

5

22

20

10

  1. The authors show that several clinicopathological characteristics are related to m6A modification genes. To exclude the influence of those features on the LASSO Cox regression model used to evaluate the prognostic value of RNA m6A regulators, the authors should train a stratified Cox regression model.

Response: Thanks for bringing up this point. The LASSO cox regression analysis was used to create a risk model for each cancer, considering factors, such as survival time, survival status, and gene expression levels. It can also be done based on the clinicopathological characteristics respectively. Due to high amount of work, we selected LUAD, as a representative tumor.

A few candidate genes were screened out considering them to be prognosis related in different clinical features, including age, gender, N, stage and T (Figure A). We divided the patients into high- and low- risk groups according to the LASSO results. Patients in the high-risk group had the worse prognosis than those in the low-risk group in every clinical feature, which is consistent with the LASSO results based on survival time, survival status, and gene expression levels (Figure B).

  1. Figure 6C shows the same information as Figure 3A and should be excluded from the manuscript.

Response: We apologize for the unclear description about Figure 3A (new Figure 4F) and Figure 6C (new Figure 4C). We have rewritten it in the revised version. In Figure 3A, we analyzed the genomic variations including 6 types, which are missense mutation, nonsense mutation, frameshift deletion, frameshift insertion, in-frame deletion and in-frame insertion. While in Figure 6C, we only analyzed the single nucleotide variations (SNVs). There is also a big difference in the nucleotide number of mutations between Figure 3A and Figure 6C. Figure 6C mainly focuses on the single nucleotide variation, whereas, Figure 3A not only focuses on the mutations of single site of nucleotide but also the multiple sites of nucleotide.

Comments on the Quality of English Language

The article would benefit from proofreading and minor editing of English.

Response: Thanks for the suggestion about the quality of English language. We have polished it with the help of a English-speaking colleague.

Round 2

Reviewer 1 Report

Comments and Suggestions for Authors

Review of

Interplay of RNA m6A modification related geneset in pan-cancer

Overall comments

The authors resubmitted the manuscript with now 9 figures down from 16.

The main criticism was that m6A genes have already been shown to be cancer-promoting. With all their impressive repertoire of bioinformatics tools the authors did not explain the biological relevance of their study in a sufficient manner. This should be addressed in the discussion. I am still expecting a summary of the data so the reader gets an overall grasp of the data and add biological context to the presented data. E.g. why are all these expression/mutations etc. of the regulators correlating with the observations/phenotypes in pan-cancer

Specific comments:

All figures: The figures are still extremely hard to read. Is it possible to move some panels to the supplement and increase the font of the remaining ones?

Especially panels, 1F, 1G, 3B, 4, 5,6,7C, 8 are not needed in the main manuscript.

Figure 2 does not really need to be shown in the main text, rather a correlation analysis with bars showing the correlation coefficient for each gene.

Author Response

Thank you very much for handling and reviewing our previous manuscript (biomedicines-3060735). We highly appreciate the comments from reviewers, which have provided great help for us to improve our work. During the past few days, we have condensed our figures to a few key ones and moved others to the supplementary,also added important ones. We have also increased the panel size and paragraphs to be more readable. We have rewritten the discussion to make it more summarized and clearer.

To our knowledge, our work sheds light on the functional roles, as well as genetic and epigenetic alterations, and prognostic value of the RNA m6A regulators in pan-cancer. It opens the door to a territory in between the two ‘hot’ fields: RNA m6A methylation and pan-cancer. We believe that many more important research will follow the important shot we fired to discover the new biomarkers or new therapies for improving the diagnosis and prognosis of cancer patients.

Therefore, we would like to resubmit our manuscript for publication consideration by Biomedicines.

Please check below for our point-by-point responses to reviewers’ comments in detail.

Point-by-point Response to Reviewers' Comments:

Reviewer 1 Round 2

Open Review

Quality of English Language

( ) I am not qualified to assess the quality of English in this paper.
( ) The English is very difficult to understand/incomprehensible.
( ) Extensive editing of English language required.
( ) Moderate editing of English language required.
( ) Minor editing of English language required.
(x) English language fine. No issues detected.

Yes

Can be improved

Must be improved

Not applicable

Does the introduction provide sufficient background and include all relevant references?

( )

(x)

( )

( )

Is the research design appropriate?

( )

(x)

( )

( )

Are the methods adequately described?

( )

( )

(x)

( )

Are the results clearly presented?

( )

( )

(x)

( )

Are the conclusions supported by the results?

( )

( )

(x)

( )

Comments and Suggestions for Authors

Review of

Interplay of RNA m6A modification related geneset in pan-cancer

Overall comments

The authors resubmitted the manuscript with now 9 figures down from 16.

The main criticism was that m6A genes have already been shown to be cancer-promoting. With all their impressive repertoire of bioinformatics tools the authors did not explain the biological relevance of their study in a sufficient manner. This should be addressed in the discussion. I am still expecting a summary of the data so the reader gets an overall grasp of the data and add biological context to the presented data. E.g. why are all these expression/mutations etc. of the regulators correlating with the observations/phenotypes in pan-cancer.

Response: Thank you for the insightful suggestion. We have rewritten the discussion and summarized the data to be clearer. To illustrate the relationship between expression/mutations etc. of the regulators and the observations/phenotypes in pan-cancer, we have added a new figure that also uses Random Forest (RF), support vector machine-recursive feature elimination (SVM-RFE) and Lasso algorithms to illustrate the m6A-related genes serving as a biomarker for prognosis.

Specific comments:

All figures: The figures are still extremely hard to read. Is it possible to move some panels to the supplement and increase the font of the remaining ones?

Especially panels, 1F, 1G, 3B, 4, 5,6,7C, 8 are not needed in the main manuscript.

Figure 2 does not really need to be shown in the main text, rather a correlation analysis with bars showing the correlation coefficient for each gene.

Response: Thank you for the constructive advice. This is indeed a very useful point. We are sorry for the smaller font of the figures. We have enhanced it and moved some panels to the supplementary figures.

We have moved the figures 1F, and 1G (i.e., figures 1E and 1F in our first round revised version) to the new supplementary figure SF1.

Figure 3B shows the relationship between the m6A modification regulators’ levels and the patients’ gender, which is a very important clinicopathological feature, such as age, grade, stage, tumor T stage, lymph node metastasis and distant metastasis. So we kept it as such, and renumbered it as figure 2B in this 2nd round revised version.

Figure 4 shows the correlation between genetic or epigenetic alterations and expression levels of m6A regulators in pan-cancer. We think this figure is important, so we only moved figure 4A to the new supplementary figure SF5A and combined figures 4a and 4b to make a new figure 3 which can be presented in one page.

Figure 5 shows the different clusters of RNA m6A regulators in TCGA pan-cancer cohort. We believe this figure is important, so we kept it and renumbered it as figure 4 in this 2nd round revised version.

Figure 6 shows the prognostic model by Lasso cox regression algorithm and is an important one. Hence, we did not move it to the supplementary, and renumbered it as figure 5 in this 2nd round revised version. Furthermore, we also used Random Forest (RF) and support vector machine-recursive feature elimination (SVM-RFE) algorithms to illustrate the m6A-related genes that serve as a biomarker for prognosis (figure 8).

Figure 7C does not exist in our first-round revised version. We moved the figures 7A and 7B to the new supplementary figure SF6.

Figure 8 shows the association of RNA m6A regulators with tumor microenvironment (TME) infiltrating cells, tumor stemness score and genomic heterogeneity in pan-cancer. This is an important figure and we have renumbered it as figure 6 in this 2nd round revised version.

We have moved the figure 2 to the supplementary figure SF3.

Submission Date

31 May 2024

Date of this review

25 Aug 2024 21:47:40

Reviewer 2 Report

Comments and Suggestions for Authors

I feel that the authors improved the quality of figures and clarified some methodical questions. I think the study benefits from reducing the number of figures and analysis, which makes the whole paper more streamlined. I have no further issues with the analysis.  

Author Response

Comments and Suggestions for Authors

I feel that the authors improved the quality of figures and clarified some methodical questions. I think the study benefits from reducing the number of figures and analysis, which makes the whole paper more streamlined. I have no further issues with the analysis.

Response: Thank you very much for handling and reviewing our manuscript. We highly appreciate your advice and encouragement which have provided great help for us to improve our work.

Submission Date

31 May 2024

Date of this review

29 Aug 2024 14:50:49

Round 3

Reviewer 1 Report

Comments and Suggestions for Authors

The authors improved the manuscript to the best of their abilities.